# HiQ-LAI: A High-Quality Reprocessed MODIS LAI Dataset with Better Spatio-temporal Consistency from 2000 to 2022

Kai Yan[1, *], Jingrui Wang[2, *], Rui Peng[2], Kai Yang[2], Xiuzhi Chen[3], Gaofei Yin[4], Jinwei Dong[5], Marie Weiss[6], Jiabin Pu[7], Ranga B. Myneni[7]

[1] Innovation Research Center of Satellite Application (IRCSA), Faculty of Geographical Science, Beijing Normal University, Beijing 100875, China

[2] School of Land Science and Techniques, China University of Geosciences, Beijing 100083, China

[3] Guangdong Province Key Laboratory for Climate Change and Natural Disaster Studies, School of Atmospheric Sciences, Sun Yat-sen University, Guangzhou, China

[4] Faculty of Geosciences and Environmental Engineering, Southwest Jiaotong University, Chengdu 610031, China

[5] Key Laboratory of Land Surface Pattern and Simulation, Institute of Geographic Sciences and Natural Resources Research, Chinese Academy of Sciences, Beijing, 100101, China

[6] Institute National de la Recherche Agronomique—Universit ́e d'Avignon et des Pays du Vaucluse (INRA-UAPV), 228 Route de l'A ́erodrome, 84914 Avignon, France

[7] Department of Earth and Environment, Boston University, Boston, MA 02215, USA

*Correspondence to*: Kai Yan, Jingrui Wang (kaiyan@bnu.edu.cn, jingruiwang@email.cugb.edu.cn)

**Abstract.** Leaf Area Index (LAI) is a crucial parameter for characterizing vegetation canopy structure and energy absorption capacity. The Moderate Resolution Imaging Spectroradiometer (MODIS) LAI has played a significant role in landmark studies due to its clear theoretical basis, extensive historical time series, extensive validation results, and open accessibility. However, MODIS LAI retrievals are calculated independently for each pixel and a specific day, resulting in high noise levels in the time series and limiting its applications in the regions of optical remote sensing is severely limited. Reprocessing MODIS LAI predominantly rely on temporal information to achieve smoother LAI profiles with little use of spatial information and may easily ignore genuine LAI anomalies. To address these problems, we designed the Spatio-Temporal Information Compositing Algorithm (STICA) for the reprocessing of MODIS LAI products. This method integrates information from multiple dimensions, including pixel quality information, spatio-temporal correlation, and original retrieval, and thus enables both "reprocessing" and "value-added data" of the existing MODIS LAI products, leading to the development of the High-Quality LAI (HiQ-LAI) dataset. Compared to ground measurements, HiQ-LAI shows better performance than the original MODIS product with Root-Mean-Square Error (RMSE) / Bias decreased from 0.87 / -0.17 to 0.78 / -0.06. This is due to the improvement of HiQ-LAI in capturing the seasonality of vegetation phenology and in reducing time-series abnormal fluctuations. The Time-series Stability (TSS) index which represents temporal stability, indicated that the area with smooth LAI time-series expanded from 31.8% (MODIS) to 78.8% (HiQ) globally, and this improvement is more obvious in equatorial regions where optical remote sensing usually cannot achieve good performance. We found that the HiQ-LAI demonstrates superior continuity and consistency compared to raw MODIS LAI from both spatial and temporal perspectives. We anticipate that the global HiQ-LAI time-series, generated by the STICA procedure on the Google Earth Engine (GEE) platform, will

substantially enhance support for diverse global LAI time-series applications. The 500 m/5 km-8 days HiQ-LAI dataset from 2000 to 2022 is available at https://doi.org/10.5281/zenodo.8296768 (Yan et al., 2023).

## 1. Introduction

In recent years, the monitoring and assessment of vegetation parameters have gained increasing importance in the context of global climate change and the emphasis on ecosystem functions (Fang et al., 2019). Leaf Area Index (LAI), which is a basic parameter affecting the processes of plant water balance, radiation absorption, and photosynthetic activity (Knyazikhin et al., 1998; Sellers et al., 1997b; Fang et al., 2019), is commonly defined as the one-sided green leaf area per unit ground horizontal surface area of broadleaf canopies and as the projected needle area of coniferous canopies (Chen and BLACK, 1992). LAI plays a crucial role in vegetation monitoring, agricultural management, and ecological modelling (Richardson et al., 2013; Zhu et al., 2016; De Wit et al., 2012). With advancements in remote sensing technology (e.g., large-scale and continuous observation) and improved data acquisition capabilities, LAI estimation and spatiotemporal dynamic monitoring have become more accurate and comprehensive (Fang et al., 2019; Ganguly et al., 2010).

Among the various time-series LAI products with global coverage, Moderate Resolution Imaging Spectroradiometer (MODIS) LAI product was among the most extensively utilized LAI datasets. MODIS LAI offers clear theoretical foundations, extensive historical time series, satisfactory validation results outcomes, and open access policy (Yan et al., 2016b, 2021a). Additionally, it does not rely on other LAI products as input data, ensuring complete independence (Myneni et al., 2002). It is often utilized as a training dataset by other products (Baret et al., 2013; Ma and Liang, 2022) and as reference data for comparison (Xiao et al., 2013; Yan et al., 2016b). The long-time series MODIS LAI dataset has made significant contributions to landmark studies on the "Greening the Earth" phenomena, the possible causes of large-scale vegetation dynamics, and the relationship between vegetation dynamics and global climate change or human activities (Mao et al., 2013; Chen and Dirmeyer, 2016; Zhu et al., 2016; Chen et al., 2019). From this perspective, the high quality of MODIS LAI product is of utmost importance.

The MODIS LAI operational algorithms comprise the main algorithm based on the three dimension radiative transfer theory and the backup algorithm relies on the empirical relationship between Normalized Difference Vegetation Index (NDVI) and canopy LAI (Myneni et al., 2002; Pu et al., 2020; Yan et al., 2018). The algorithm utilizes a Look-Up Table (LUT) inversion strategy and introduces the biome classification map as prior knowledge to reduce uncertainties associated with ill-posed inversion problems. By modelling the photon transfer process, the surface spectral Bidirectional Reflectance Factors (BRFs) are linked to structural and spectral parameters of the vegetation canopy and soil. Based on atmospherically corrected BRFs and their uncertainties, the algorithm identifies candidate LAIs by comparing observed and modelled BRFs stored in biome-specific LUTs (Knyazikhin et al., 1998; Knyazikhin, 1999). When the uncertainty of the input BRFs falls within a point on the red-NIR plane and an area, all canopy or soil patterns are considered as the candidate solutions, and the mean LAI values of these solutions are used as the output values of the main algorithm. The backup algorithm is triggered when the main

algorithm fails, such as when the uncertainty of input BRFs exceeds a threshold or when there are inaccuracies in BRFs simulation due to deficiencies in the radiative transfer model. The best retrievals are then selected using the temporal compositing method, and the 4-day or 8-day product is generated from the daily retrievals. Therefore, MODIS LAI retrievals are calculated independently for each pixel and daily. Differences in adjacent observation conditions lead to significant uncertainty in the LAI time series. Specifically, atmospheric conditions (e.g., cloud cover, snow, and aerosol pollution), sensor malfunctions, and the inherent uncertainties of the retrieval algorithm all introduce challenges, resulting in poor spatiotemporal consistency and high noise in MODIS LAI products (Brown et al., 2020; Fuster et al., 2020; Yan et al., 2021b). Consequently, inconsistency and excessive noise impose limitations on its practical applications in research involving yield estimation, crop-growth monitoring, terrestrial carbon monitoring, and global ecosystem dynamic simulation (Li et al., 2017; Xiao et al., 2009; Chen et al., 2020).

Many methods have been proposed to reprocess MODIS LAI products (ranging from C4 to C6) to improve their quality. A pressing need exists for continuous, high-quality, and easily accessible LAI datasets to better facilitate investigations in land surface process simulation, climate modelling, and global change research, Fang et al. (2006) proposed a Spatio Temporal Filtering (TSF) method that integrates multi-seasonal average trend (background) and seasonal observations to generate spatially and temporally continuous MODIS LAI C4 products for the North American region to fill in and improve the gaps and poor quality values caused by cloud cover, seasonal snow cover, and instrument problems. Gao et al. (2008) utilized TIMESAT to process MODIS LAI C4 product, aiming to fit the LAI time series profile by incorporating sufficient high-quality data and replacing low-quality or missing observations, thus obtaining high-quality spatiotemporal continuous LAI time series for the North America region to produce temporally smoothed and spatially continuous biophysical data for the North American Carbon Program. Aiming to generate continuous input dataset for global climate models, Yuan et al. (2011) improved Fang's method and proposed the modified TSF (mTSF) method to conduct simple data assimilation for relatively low-quality data and used post-processing TIMESAT and SG filtering to obtain the final improved MODIS LAI C5 product. Recently, Yuan's group (Lin et al., 2023) reprocessed MODIS LAI C6 products using a similar procedure. While these methods effectively utilize temporal and QC layers information, they frequently overlook the utilization of spatial information or rely on spatial correlation as an alternative and place a greater emphasis on leveraging temporal information. Consequently, although the LAI profile may appear smoother, genuine land surface LAI anomalies (e.g., caused by forests fire) may be artificially removed.

To address this issue, we proposed a Spatio-Temporal Information Compositing Algorithm (STICA) in a previous study (Wang et al., 2023). This algorithm directly introduces prior Multiple Quality Assessment (MQA) information and spatiotemporal correlation information into the MODIS LAI C6.1 product. Firstly, we carefully assessed the quality of MODIS LAI retrievals to obtain MQA information. Subsequently, the quality, spatiotemporal information, and relative original observation records are fully utilized, and these pieces of information are weighted and averaged according to our fusion strategy. More robust results are obtained by considering multiple dimensions of information to compensate for the limitations of using a single information source and by preserving as real LAI anomalies as possible. The advantages of our approach are

as follows: 1) ensure consistency with existing MODIS LAI products while preserving the unique benefits of MODIS that maintain the original physics-based (Radiative Transfer Model, RTM) production process, and 2) leveraging pixel quality information to improve MODIS LAI retrievals with poor quality, facilitating the "reprocessing" and "value-added data" of the existing product. We anticipate that the High-Quality LAI (HiQ-LAI) product effectively addresses regions with quality issues

while maintaining good consistency with the original MODIS product, holding significant implications for other LAI/FPAR products development.

We implemented the entire algorithm process using the Google Earth Engine (GEE) cloud computing platform (Gorelick et al., 2017) to reprocess MODIS C6.1 LAI, resulting in the production of the HiQ-LAI dataset covering the years 2000 to 2022 on a global scale. The accuracy of HiQ-LAI was evaluated through ground-based validation and compared with MODIS

LAI at a global scale and across different biome types. The temporal consistency and trends of global LAI products were analysed, with in-depth comparative assessments conducted for regions exhibiting significant quality issues.

## 2. Materials

### 2.1 MODIS Land Cover map: MCD12Q1

The biome classification map serves as an auxiliary dataset for MODIS LAI, primarily aimed at reducing uncertainty in

the retrieval algorithm. In the MODIS retrieval algorithm, parameters are configured based on the biome classification map to establish an accurate relationship between satellite observations and ground parameters (Knyazikhin, 1999). The MODIS land cover type product (MCD12Q1) provides global land cover maps at annual time steps and 500 m spatial resolution from 2001 to the present. The product was created using supervised classification of MODIS reflectance data for a total of 13 scientific datasets. The MCD12Q1 adopts LAI legacy classification scheme, including B1 (grass and cereal crops); B2 (shrub); B3

(broadleaf crops); B4 (savanna); B5 (evergreen broadleaf forest, EBF); B6 (deciduous broadleaf forest, DBF); B7: (evergreen coniferous forest, ENF); B8 (deciduous coniferous forest, DNF) (Yan et al., 2016a; Sulla-Menashe and Friedl, 2018). Fig. 1 illustrates the approximate global distribution of biome types based on the land cover in 2021.

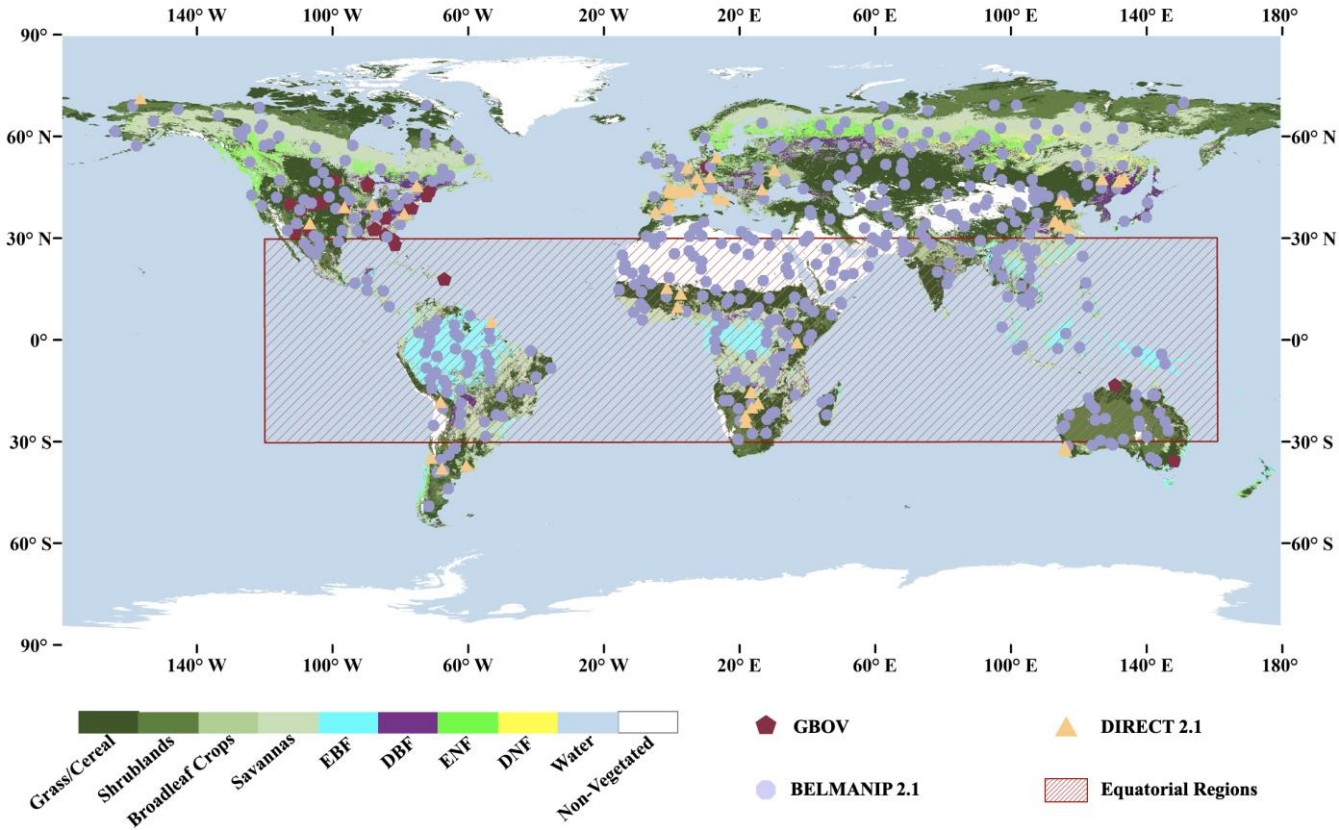

**Figure 1. Geographical distribution of the selected sites. The background color indicates the biome types of the 2021 MCD12Q1 classification scheme. The red hexagon, yellow triangle, pink dots, and red frame represent the GBOV sites, DIRECT 2.1 sites, BELMANIP 2.1 sites, and Equatorial Regions, respectively.**

## 2.2 MODIS LAI/FPAR product: MOD15A2H

The standard MODIS LAI C6.1 product suite (MOD15A2H) offers global coverage with an 8-day temporal resolution

and a 500-m spatial resolution (Yan et al., 2016a) spanning from February 2000 to the present. Typically, there are 46

composites per year, but some composites may be missing due to factors such as sensor issues or other anomalies (e.g., in

2001, 2016, and 2022). This LAI product is projected onto a sinusoidal grid and distributed as a standard Hierarchical Data

Format (HDF) file. Each file contains six Science Data Sets (SDSs): Fraction of Photosynthetically Active Radiation absorbed

by vegetation (FPAR), LAI, FparLai_QC, FparExtra_QC, FparStdDev, and LaiStdDev. The LAI, FparLai_QC ,and LaiStdDev

layers store the LAI retrieval, quality control information, and retrieval uncertainty, respectively (Knyazikhin et al., 1998; Yan

et al., 2016a; Myneni, 2020). Therefore, for this study, we utilized these three SDSs.

**2.3 Ground LAI reference**

With the increasing availability of Earth observation products, increasing attention has been paid to the product uncertainty assessed by verification based on ground measurements (Baret et al., 2006; Fang et al., 2012; Wenze et al., 2006). To evaluate the performance of the HiQ-LAI product, we utilized the Copernicus Ground Observation Validation (GBOV) LAI, DIRECT 2.1 LAI, and BELMANIP 2.1 as ground site references (Bai et al., 2019; Baret et al., 2006; Morisette et al., 2006).

**2.3.1 GBOV**

The Copernicus Ground-Based Observations for Validation (GBOV) service, which is part of the Copernicus Global Land Service (CGLS), is dedicated to the development and distribution of robust in situ datasets from various ground monitoring sites for the systematic and quantitative validation of land products (Bai et al., 2019; Brown et al., 2020). A comprehensive GBOV reference measurement database has been established through quality control and raw measurements reprocessing obtained from existing in situ sites. This database includes canopy reflectance, surface albedo, LAI, FPAR, cover area, 5 cm soil moisture, and surface temperature. Currently, 29 available sites provide LAI references from 2013 to 2022. The data from this database are freely accessible to the scientific community through the GBOV portal (https://gbov.acri.fr). In this study, we used the Land Products 3- leaf area index (LP3) as reference LAI.

**2.3.2 DIRECT V2.1**

We also employed the DIRECT V2.1 ground measurement reference to validate satellite-based products. This validation dataset was curated from a collection of sites that followed the CEOS/WGCV-LPV guidelines for collecting and processing ground survey data (Morisette et al., 2006; Garrigues et al., 2008). Mean values of LAI, the maximum fraction of photosynthetically active radiation absorbed by vegetation (FPAR), and vegetation cover (FCover) over a 3km × 3km area were compiled from the DIRECT V2.1 database. By the CEOS WGCV LPV LAI good practices (Fernandes et al., 2014), the ground data were upscaled using an empirical "transfer function" between high spatial resolution radiation data and biophysical measurements to appropriately account for the spatial heterogeneity of the site. The DIRECT V2.1 database constitutes a major effort of the international community to provide ground references for the validation of LAI and FAPAR ECVs. It currently encompasses 176 sites across seven major biome types around the world, with 280 LAI values, 128 FPAR, and 122 FCOVER values spanning from 2000 to 2021 (https://calvalportal.ceos.org/lpv-direct-v2.1).

**2.4 BELMANIP V2.1**

The BELMANIP site network was designed to represent the global variability of vegetation types and climatological conditions, independent of ground-based experimental measurements (Baret et al., 2006). The network primarily incorporates sites from existing experimental networks (FLUXNET, AERONET, VALERI, BigFoot, etc.) supplemented by sites from the

GLC2000 land cover map. The BELMANIP V2.1 we used was constructed using the GLOBCOVER vegetation land cover map derived from MERIS images in 2009. Site selection was conducted for each band of latitude (10° width), ensuring that the selected sites maintained the same proportion representation of biome types as the entire latitude band. These sites are characterized by a uniform and nearly flat topography within a 10×10 km² area, with a minimum proportion of urban areas and permanent water bodies. The updated BELMANIP V2.1 dataset (Fig. 1) added 25 sites corresponding to bare land (desert) and tropical forests, resulting in a total of 445 sites (https://calvalportal.ceos.org/web/olive/site-description). It is important to note that most of these sites do not have ground reference measurements. Consequently, the network is primarily used for comparative analysis among sites rather than direct validation.

## 3. Method

### 3.1 Proposed Spatio-temporal Information Compositing Algorithm

We proposed a Spatio-Temporal Information Composition Algorithm (STICA) aimed at reducing noise fluctuations and improving the overall quality of the MODIS LAI product. This algorithm directly incorporates the prior spatiotemporal correlation information and Multiple Quality Assessment (MQA) information into the existing MODIS LAI product. The detailed algorithmic process can be found in the article published by Wang et al. (2023). The algorithm consists of four main steps: multiple quality assessment, employing spatial correlation information, employing temporal correlation information, and multiple information compositing.

Satellite remote sensing observations are often subject to uncertainties arising from climatic factors, sensor malfunctions, and other sources, resulting in varying levels of uncertainty for individual pixels. To address this issue, this approach employed multiple indicators to evaluate the uncertainty for each pixel (referred to as MQA hereafter). These indicators encompass the algorithm path, STD LAI, and Relative Time-series Stability (TSS). The algorithm path (AP) is a crucial quality index, distinguishing between the main and backup algorithms. The main algorithm offers superior quality and precision retrieval, and the weight ratio of the main algorithm and backup algorithm is determined as 6:4 in the previous study (Wang et al., 2023). STD LAI reflects the retrieval uncertainty. The AP and STD LAI are derived from the FparLai_QC and LaiStdDev layers of the original MODIS data. The third indicator, Relative TSS (RE-TSS), indicates the fluctuation of a time series (Zou et al., 2022). Following the principle of assigning a higher weight to smaller values, STD LAI and RE-TSS are incorporated into the retrieval with the main algorithm, resulting in the generation of a new quality classification indicator, MQA. Subsequently, the Inverse Distance Weighting (IDW) method is utilized on the spatial scale to calculate the weighted average of all eligible pixels (belonging to the same land cover type) within the half-width of 4 pixels and the power exponent of 2 (Wang et al., 2023) of the target pixel. In this algorithm, the contribution of a pixel is determined not only by its spatial distance but also by its MQA value. In a word, pixels with closer proximity and higher MQA value make a more significant contribution to the target pixel. On the temporal scale, the Simple Exponential Smoothing (SES) method is employed to calculate the weighted average of all eligible pixels within the smoothing parameter of 0.5 and the half-length of 3 (Wang et al., 2023). Pixels that

are closer in time to the target pixel and possess higher MQA values are assigned greater weights. Utilizing spatial/temporal correlation is based on spatial and temporal autocorrelation, i.e., everything is related to everything else, but near things are more related than distant things. The final step of the algorithm is to take a weighted average of the original MODIS LAI and the LAI calculated using spatial/temporal correlation, with their respective weights quantified using an indicator (TSS) that represents the temporal fluctuation of the time series (Zou et al., 2022). All processes of the method are implemented using the GEE cloud computing platform. The reprocessed LAI dataset, namely the HiQ-LAI product, has been generated with the help of the powerful cloud computing capability of GEE, covering the period from 2000 to 2022.

## 3.2 Assessment of LAI Datasets

In this study, we utilized the GBOV LAI measurements from a total of 29 sites spanning from 2013 to 2021 as our ground reference LAI (Bai et al., 2019; Brown et al., 2020). A 3 km × 3 km square centered on the site location was selected as the study area (Fig. 1) so that the corresponding LAI product of each site was 36 (6 × 6) pixels. To enhance the credibility of the ground truth LAI, we filtered the ground LAI reference of these 29 sites based on the criterion that the "effective pixel" exceeded 90% and the input and output of land product value in the data aggregation process were within the specified range. This filtering process yielded a total of 818 reliable verification data points. Contrary to previous studies (Wang et al., 2023) that utilized only 2018 data from the GBOV site as a reference, this study expanded the timeline from 2013 to 2021, increased the number of sites from 24 to 29, and raised the criterion for effective pixels from 80% to 90%. These modifications were aimed at enhancing the reliability of the ground LAI data. Additionally, previous research focused on proposing and testing algorithms mainly at the tile scale, but this study migrated the algorithm to GEE for generating global long-term data series. Furthermore, the scope of analysis was also broadened to a global spatial scale and long-term time series. A comparative analysis was conducted at the spatial scale to examine the global spatial distribution of LAI in February and July 2021. The mean LAI values for latitude bands were then calculated at 1-degree intervals during these specific months. Furthermore, we compared the global consistency of MODIS LAI and HiQ-LAI in 2021 using the BELMANIP V2.1 sites (445 in total) (Baret et al., 2006). Employing these sites not only reduced the computational burden on a global scale but also mitigated additional uncertainties arising from geometric registration bias and land cover misclassification. Similar to the GBOV, we selected a study area of 6 × 6 pixels centered on each site location (Fig. 1). The MCD12Q1 data in 2021 were utilized to determine the biome type of each site, which was further classified into pure pixels and hybrid pixels based on B1 - B7. The total amount of data available for comparison was 16420. Additionally, we used DIRECT V2.1 ground measurements in this research (Morisette et al., 2006; Garrigues et al., 2008). However, these data were not utilized for direct validation due to the discontinuity in the observed time series at these sites. Instead, the DIRECT V2.1 sites provided valuable reference values in Sect. 5.2. Similarly, a research area of 6 × 6 pixels was selected for each site, and we compared the $R^2$ and RMSE of the two products with sites across different quality grades. The analysis involved determining the RMSE reduction percentage and $R^2$ increase in the percentage of HiQ-LAI relative to MODIS under various quality grades.

In this study, the Theil-Sen's slope (TS) method and Mann-Kendall (MK) test (Suhartati, 2013; Theil, 1992) were employed to extract LAI trends from the two products. The TS method computes pairwise slopes across the study period, with the median slope representing the sign and magnitude of the long-term trend. Unlike ordinary least-square linear regression, the TS trend is less susceptible to the influence of outliers. Meanwhile, the MK test is utilized to determine the significance of the trend (Kendall, 1948). The combination of TS and MK forms a robust approach for identifying trends in long-term sequential data. TS and MK are calculated as follows:

$$TS = median\left(\frac{X_j - X_i}{j - i}\right), \qquad 2000 \leq i < j \leq 2022 \tag{1}$$

where $X_j$ and $X_i$ represent the LAI value of year $j$ and year $i$, respectively. If $TS > 0$ indicates an increasing trend, while indicates a decreasing trend. Following this, the MK test was applied to assess the annual mean trends for MODIS LAI and HiQ-LAI from 2000 to 2022, ensuring the statistical significance of the identified trends.

$$S = \sum_{i=1}^{n-1} \sum_{j=i+1}^{n} sgn(x_j - x_i) \tag{2}$$

$$Var(S) = \frac{n(n-1)(2n+5) - \sum_{i=1}^{m} t_i(t_i - 1)(2t_i + 5)}{18} \tag{3}$$

$$Z_s = \begin{cases} \dfrac{S - 1}{\sqrt{Var(S)}}, & \text{if } S > 0 \\ 0, & \text{if } S = 0 \\ \dfrac{S + 1}{\sqrt{Var(S)}}, & \text{if } S < 0 \end{cases} \tag{4}$$

where $S$ represents the sum of step function values obtained from the differences between any two distinct points within the time series, $n$ signifies the total number of data points, $m$ indicates the count of continuous groups in the data (duplicate data set), and $t_i$ refers to the associated count (the number of repetitions in the ith range). Ultimately, we calculate the test statistic $Z_s$, when $|Z_s| > Z_{1-\alpha/2}$ means reject the null hypothesis (i.e., the absence of a trend), with $\alpha$ representing the significance level. In our analysis, we set $\alpha = 0.05$, with $Z_{1-\alpha/2} = 1.96$ (indicating significance at 90% and 95% confidence levels when equal to 1.65 or 1.96, respectively).

The definition of TSS as follows:

$$TSS(t) = \frac{\left| \begin{array}{l} \left(X(t_{n+1}) - X(t_{n-1})\right) \times t_n - X(t_n) \times (t_{n+1} - t_{n-1}) - \\ \left(X(t_{n+1}) - X(t_{n-1})\right) \times t_{n-1} + X(t_{n-1}) \times (t_{n+1} - t_{n-1}) \end{array} \right|}{\sqrt{\left(X(t_{n+1}) - X(t_{n-1})\right)^2 - (t_{n+1} - t_{n-1})^2}} \tag{5}$$

where $X(t_n)$, $X(t_{n+1})$, and $X(t_{n-1})$ represent the LAI value at target moment t, the adjacent time series data obtained at the previous moment, and the next moment, respectively. The TSS denotes the deviation of a value at a given point in time from the linear interpolation line. In this study, higher TSS values indicate greater variability over time.

## 4. Results and Discussion

### 4.1 Validation based on Ground LAI Reference

Figure 2 depicts the validation results obtained from GBOV ground reference LAI, highlighting the superior performance of HiQ-LAI compared to MODIS LAI. From MODIS LAI to HiQ-LAI, $R^2$ increases from 0.69 to 0.71, RMSE decreases from 0.87 to 0.78, Relative RMSE (RRMSE) decreases from 26.63% to 24.04%, and Bias shifted from $-0.17$ to $-0.06$. Notably, the fitted line of MODIS LAI deviated more prominently from the 1:1 line compared to HiQ-LAI, indicating that HiQ-LAI exhibits higher accuracy with the ground reference LAI. The two methods had negligible differences in LAI performance in terms of pure grasslands and mixed grasses biome types. However, MODIS LAI exhibited substantial overestimation for pure forest type (triangle). Incorporating data quality information (MQA), we observed a correspondence between high LAI values and low MQA values, and the majority of pure forests and mixed forests sites show the phenomenon of low MQA values. In contrast, the mixed savannas and forests biome (pentagram) displayed underestimation in MODIS LAI. Furthermore, MODIS LAI exhibited noticeable abnormal retrieval values (red pentagram in the upper left corner), whereas HiQ-LAI effectively mitigated these issues. Comparing the LAI difference distribution among various vegetation types (Fig. 3) revealed that HiQ-LAI exhibited a tighter concentration around the zero value, resulting in decreased RMSE across most categories, except for the third biome type (Grasses & Shrubs). Mixed savannas and forests emerged as the vegetation types with the widest MODIS LAI difference range. The enhanced HiQ-LAI notably narrowed this distribution range, although the median and mean deviated further from zero. Notably, the two biome types exhibiting the most conspicuous changes in difference distribution were pure forest and mixed crops and savannas. The verification analysis (Fig. 4), comparing both products against GBOV LAI references across different seasons, demonstrated that HiQ-LAI had superior performance over MODIS LAI throughout all four seasons and exhibited outperformance with the ground references. Analyzing the LAI density distribution revealed that MODIS LAI (green) skewed towards higher values on the right side compared to HiQ-LAI (black). This indicated that MODIS LAI predominantly occupied high-value areas. Furthermore, the RMSE and RRMSE of HiQ-LAI are always smaller than that of MODIS LAI.

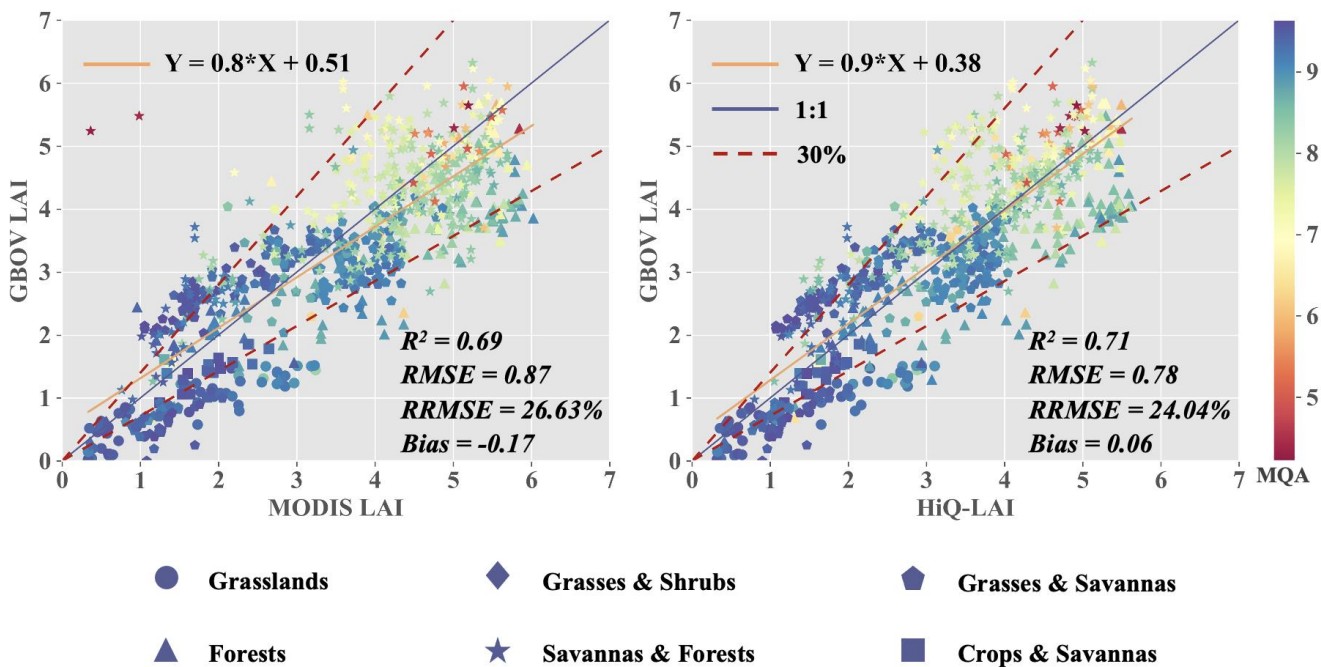

Figure 2. Scatter plots comparing MODIS LAI (left) and HiQ-LAI (right) with GBOV LAI reference (29 sites and 818 measurements). Symbol colors correspond to MQA from 4 (poor) to 10 (good), and the shape of characters represents different biome types.

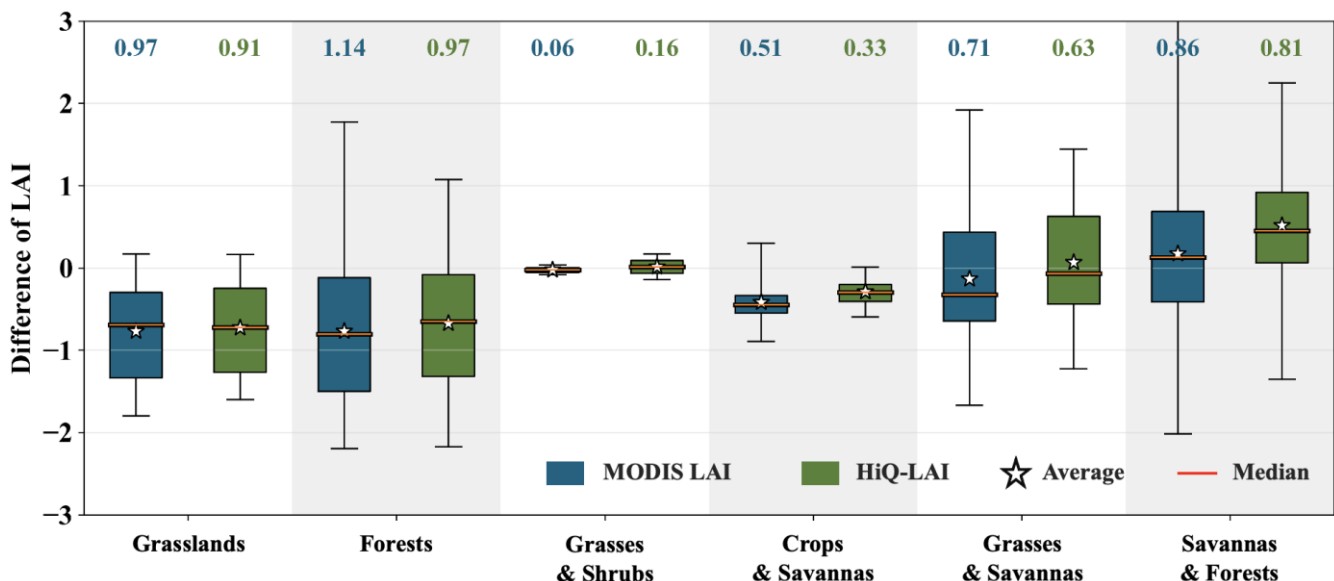

Figure 3. Accuracy comparison between two products and GBOV LAI under different vegetation types. The numbers at the top represent the RMSE between the two products and the GBOV LAI reference, respectively.

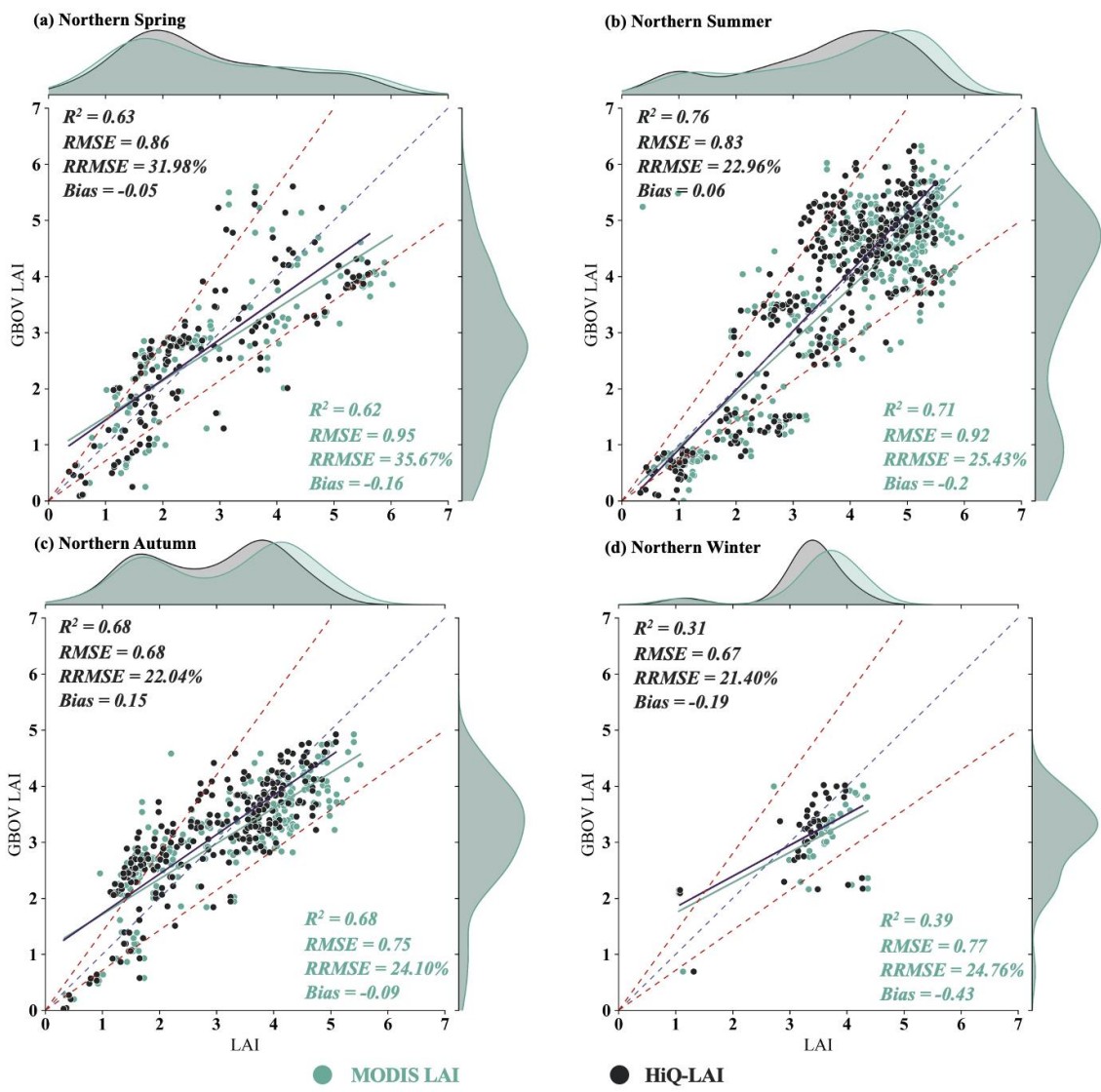

**Figure 4. Scatter plot distribution comparison MODIS LAI (green) and HiQ-LAI (black) with GBOV LAI reference under different seasons. Northern Spring, Summer, Autumn, and Winter include the months of March to May, June to August, September to November, and December to February, respectively.**

The LAI curve should exhibit stable annual and seasonal fluctuations in response to seasonal changes. Nevertheless, various factors such as harsh atmospheric conditions, sensor hardware issues, and other technical challenges introduce significant discrepancies between adjacent time windows of MODIS LAI observations (Garrigues et al., 2008), leading to abnormal fluctuations in the LAI time series profiles. As depicted in the LAI time series comparison diagram (Fig. 5 and Fig. S1), HiQ-LAI demonstrates improved quality for different vegetation types. The time series curves generated by HiQ-LAI

exhibit reduced abnormal fluctuations and better alignment with expected phenological patterns compared to the original MODIS LAI. Notably, the RMSE of all sites (totalling 29, as shown in Table 1, Fig. 5, and Fig. S1) exhibit varying degrees

of reduction, except for the DSNY, JERC, JORN, and SERC sites. Four sites (MOAB, STER, VASN, WOMB) were excluded from calculating correlation coefficients due to the unavailability of validation data after filtering. Note that since in-situ measurements may be sensitive to all elements of the canopy, the resulting estimate should technically be called the term plant

area index (PAI) (Brown et al., 2021). Studies have demonstrated that PAI may overestimate LAI by as much as 61% in certain scenarios (Brown et al., 2021). Considering this, Fig. 2, Fig. 5 and Fig. S1 illustrate that MODIS overestimates the LAI of forests to some extent, a finding consistent with prior research (Yan et al., 2016b, 2021a). However, HiQ-LAI corrects this overestimation to a certain degree.

**Table 1.** Comparison of MODIS LAI and HiQ-LAI over GBOV sites

| Biome Type | Site | M_RMSE | H_RMSE | M_R$^2$ | H_R$^2$ | M_RRMSE (%) | H_RRMSE (%) | M_Bias | H_Bias |
|---|---|---|---|---|---|---|---|---|---|
| | **CPER** | 0.44 | 0.40 | 0.21 | 0.22 | 75.90 | 69.32 | -0.39 | -0.37 |
| | **KONA** | 1.26 | 1.19 | 0.47 | 0.68 | 102.32 | 96.31 | -1.20 | -1.16 |
| | **MOAB** | --- | --- | --- | --- | --- | --- | --- | --- |
| **Grasslands** | **ONAQ** | 0.31 | 0.29 | 0.06 | 0.04 | 95.57 | 90.81 | -0.20 | -0.19 |
| | **SRER** | 0.21 | 0.21 | 0.84 | 0.86 | 45.16 | 44.94 | -0.11 | -0.05 |
| | **STER** | --- | --- | --- | --- | --- | --- | --- | --- |
| | **WOOD** | 1.29 | 1.19 | 0.55 | 0.66 | 110.19 | 102.07 | -1.24 | -1.15 |
| | **HARV** | 0.82 | 0.53 | 0.72 | 0.83 | 19.37 | 12.41 | -0.24 | -0.15 |
| **Forests** | **TALL** | 1.29 | 1.14 | 0.77 | 0.81 | 38.72 | 34.10 | -1.08 | -0.96 |
| | **TUMB** | 1.33 | 1.31 | 0.94 | 0.95 | 77.58 | 76.23 | -1.19 | -1.19 |
| **Grasses** | **JORN** | 0.06 | 0.16 | 1.00 | 1.00 | 14.95 | 35.91 | -0.03 | 0.01 |
| **& Shrubs** | **VASN** | --- | --- | --- | --- | --- | --- | --- | --- |
| **Crops** | **BLAN** | 0.53 | 0.34 | 0.41 | 0.71 | 36.72 | 23.76 | -0.44 | -0.31 |
| **& Savannas** | **LAJA** | 0.32 | 0.17 | 0.99 | 0.20 | 25.31 | 13.42 | -0.24 | -0.13 |
| | **GUAN** | 0.72 | 0.49 | 0.36 | 0.43 | 22.23 | 15.13 | -0.59 | -0.33 |
| **Grasses** | **JERC** | 0.72 | 0.80 | 0.70 | 0.81 | 24.69 | 27.42 | 0.61 | 0.76 |
| **& Savannas** | **LITC** | 0.85 | 0.68 | 0.58 | 0.66 | 149.71 | 120.21 | -0.83 | -0.66 |
| | **NIWO** | 0.49 | 0.31 | 0.68 | 0.64 | 66.89 | 42.19 | -0.47 | -0.30 |
| | **BART** | 0.75 | 0.46 | 0.80 | 0.89 | 19.60 | 11.87 | -0.14 | 0.06 |
| | **DELA** | 1.16 | 1.10 | 0.05 | 0.03 | 26.67 | 25.17 | 0.42 | 0.91 |
| | **DSNY** | 0.76 | 0.89 | 0.75 | 0.86 | 30.34 | 35.54 | 0.75 | 0.89 |
| | **HAIN** | 1.11 | 0.45 | 0.50 | 0.88 | 24.83 | 10.16 | 0.16 | 0.21 |
| | **ORNL** | 0.73 | 0.68 | 0.44 | 0.55 | 18.50 | 17.05 | -0.02 | 0.39 |
| **Savannas** | **OSBS** | 0.45 | 0.42 | 0.85 | 0.74 | 18.44 | 17.24 | 0.20 | 0.30 |
| **& Forests** | **SCBI** | 1.05 | 0.81 | 0.52 | 0.78 | 23.00 | 17.67 | 0.41 | 0.54 |
| | **SERC** | 0.84 | 1.38 | 0.83 | 0.80 | 18.85 | 30.92 | 0.75 | 1.31 |
| | **STEI** | 0.75 | 0.65 | 0.54 | 0.67 | 18.17 | 15.81 | -0.20 | 0.42 |
| | **UNDE** | 0.42 | 0.28 | 0.03 | 0.47 | 9.50 | 6.40 | 0.07 | 0.14 |
| | **WOMB** | --- | --- | --- | --- | --- | --- | --- | --- |


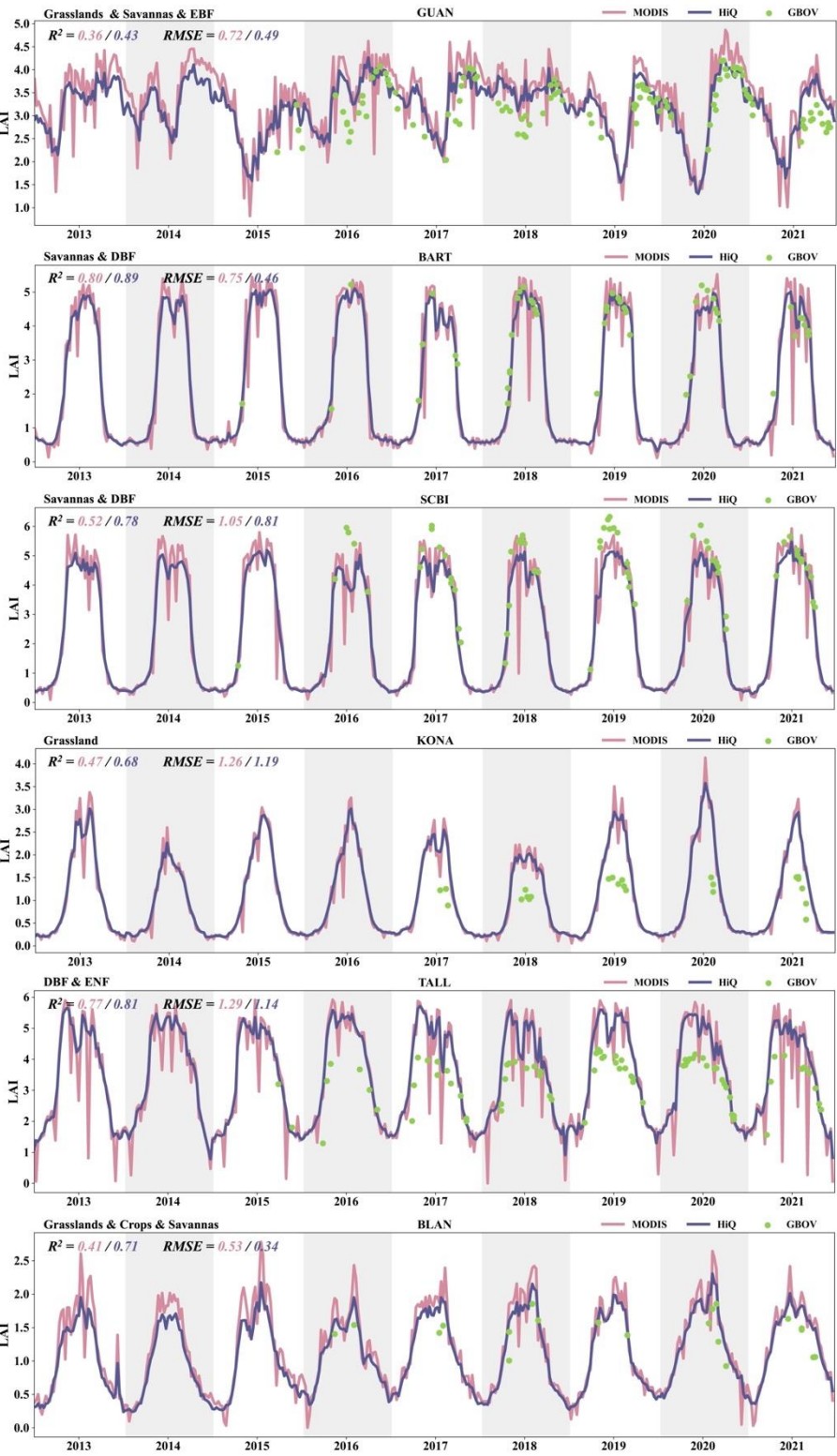

**Figure 5. Comparison of LAI time series from 2013 to 2021 at six GBOV sites representing different biome types. The LAI values from MODIS C6 product and the newly proposed HiQ-LAI are shown as solid pink and purple lines, respectively. The LAI values from ground-based GBOV reference are denoted green dots.**


## 4.2 Comparison of Global Spatial Distribution

Figure 6 displays the 5km spatial distribution of global MODIS LAI and HiQ-LAI for February and July. Overall, the two products demonstrate comparable spatial patterns across global regions. However, notable distinctions are shown in February, particularly in the Amazon Forest region of South America. The mean LAI latitudinal profile for both products show
remarkable similarity, indicating consistent overall performance. Nevertheless, it is worth noting that the standard deviation of HiQ-LAI (green shading in Fig. 6 (c1) and (c2)) consistently falls within the range of the standard deviation of MODIS LAI (red shading in Fig. 6 (c1) and (c2)). This suggests that HiQ-LAI exhibits more stability at a global scale, indirectly implying superior data quality and enhanced stability compared to MODIS LAI.

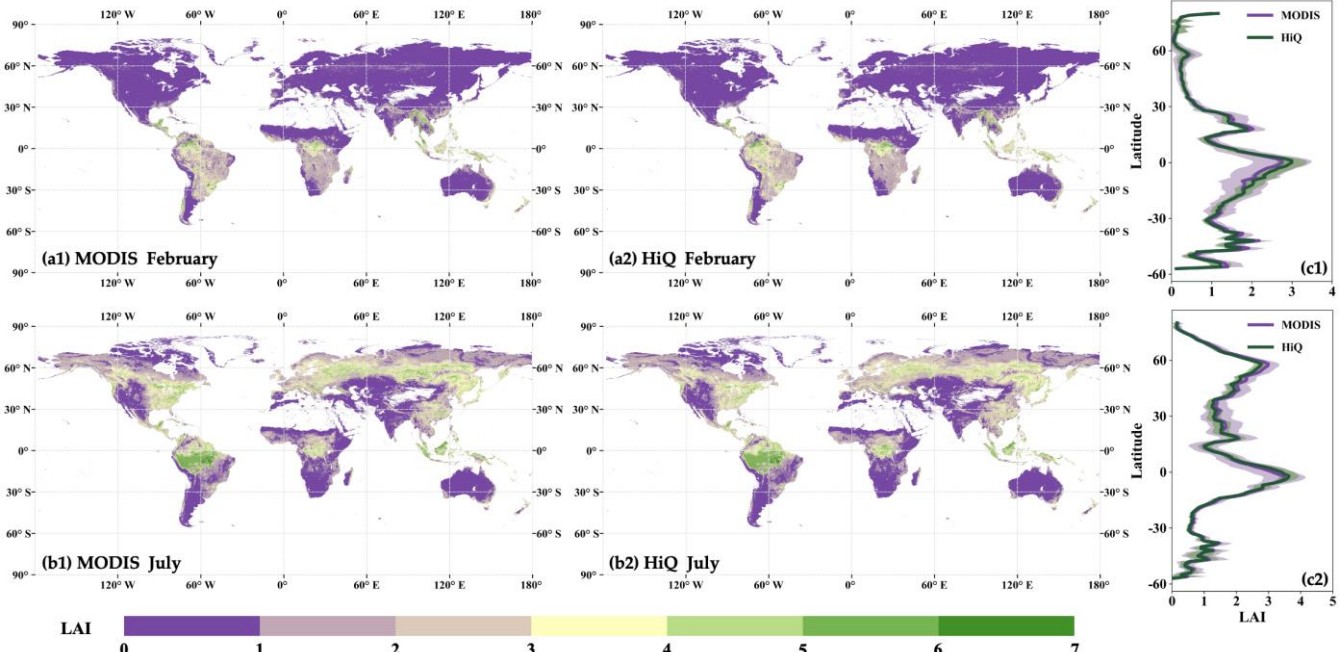

**Figure 6. Globally spatial distribution of the mean MODIS LAI (a1) and HiQ-LAI (a2) values and latitudinal profile distribution (c1) for February 2021. (b1), (b2) and (c2) are same with (a1), (a2) and (c1), respectively, but for July 2021. In (c1) and (c2), the latitude interval is 1°, the red and green line denoted MODIS LAI and HiQ-LAI, respectively.**

## 4.3 Biome Specific Comparison over Global Scale

To further assess the performance of HiQ-LAI across different biome types, we extracted MODIS LAI and HiQ-LAI
values from the BELMANIP V2.1 site. A scatter plot (Fig. 7) was constructed based on the biome type to evaluate the

consistency between the two products. The result demonstrates that, except for B5, the $R^2$ for other pure vegetation types exceeds 0.88, and B1 and B3 surpassed 0.95. The consistency of mixed pixels is also relatively high, as indicated by an RMSE of 0.42 and an $R^2$ of 0.86. However, B5 exhibits a significant disparity, with an $R^2$ value of 0.15. The Biome of B5 is primarily grown in the Amazon rainforest region of South America, Indonesia, and central Africa (Fig. 1). These regions are affected by the long-term influence of large cloud cover, high concentration of aerosol and saturation of Red-NIR, resulting in limited availability of high-quality observation data and poor accuracy in LAI retrieval (Xu et al., 2018; Yan et al., 2016b). Consequently, it can be inferred that HiQ-LAI and MODIS LAI are consistent in most areas (different vegetation types) but inconsistent in local areas that suffer from quality criticism.

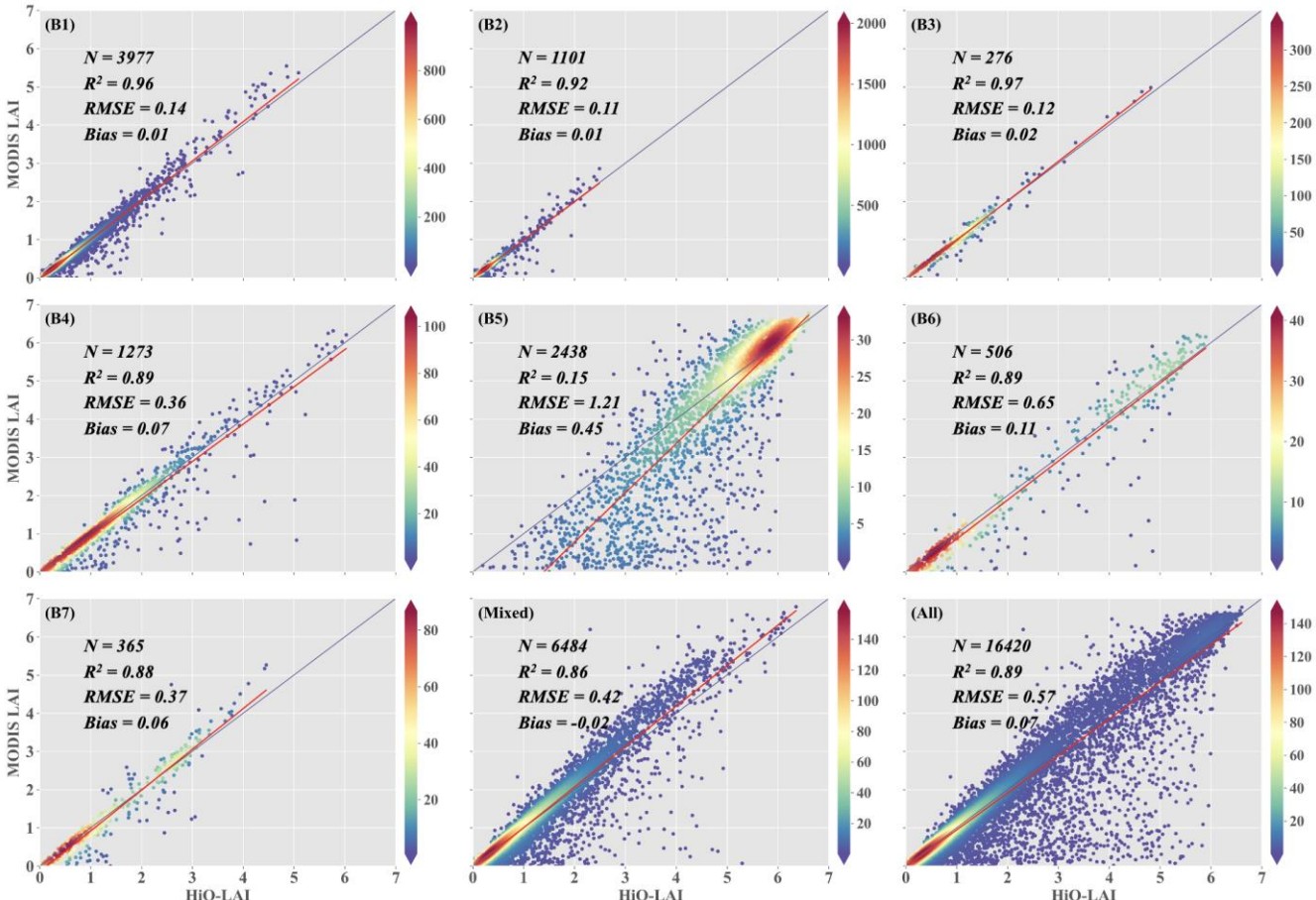

Figure 7. Density scatter plots comparison of MODIS LAI and HiQ-LAI in 2021 using the BELMANIP V2.1 sites (445 sites). B1: grass and cereal crops, B2: shrub, B3: broadleaf crops, B4: savanna, B5: evergreen broadleaf forest, B6: deciduous broadleaf forest, B7: evergreen coniferous forest, B8: deciduous coniferous forest.

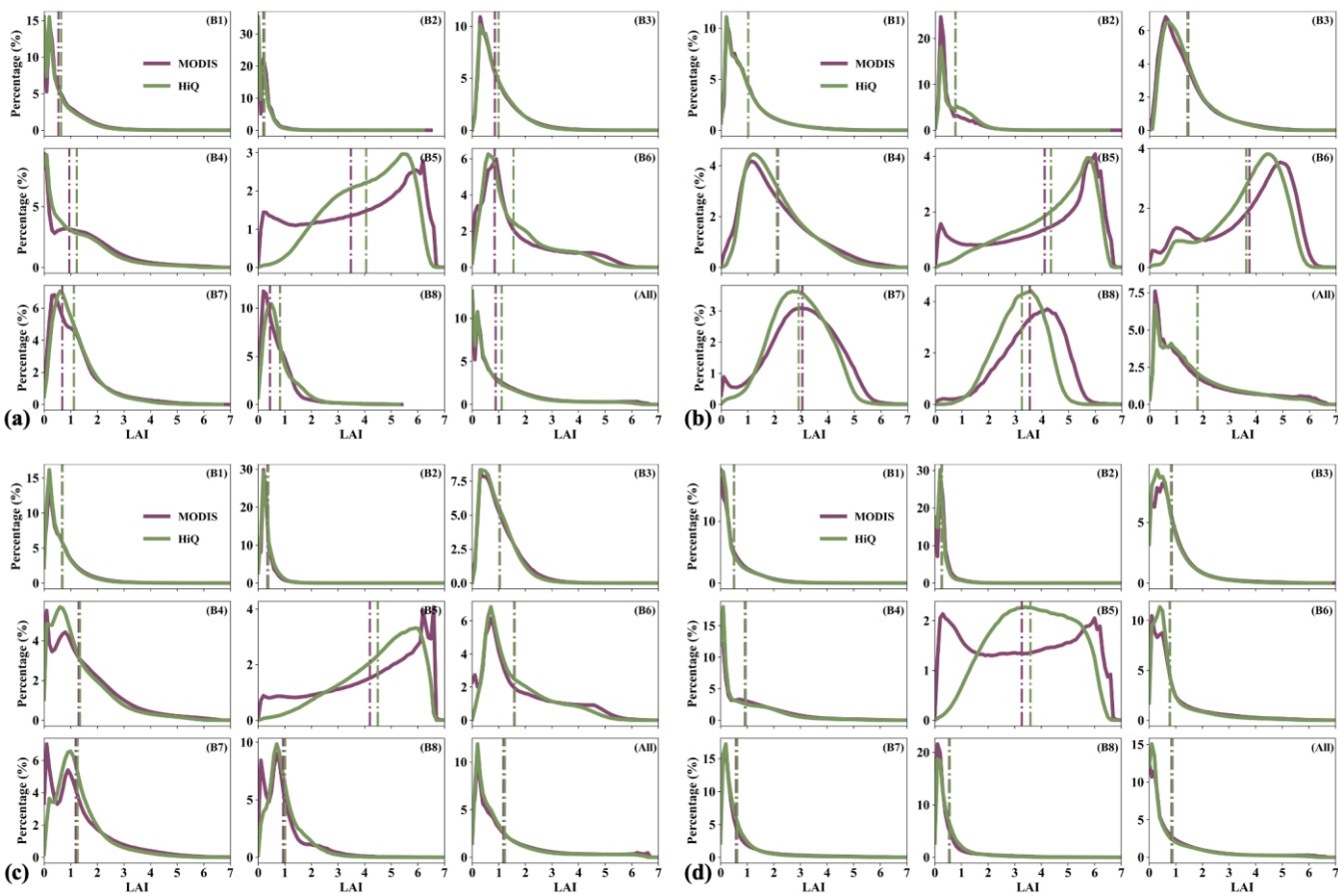

Figure 8. The numerical distribution ranges of MODIS LAI and HiQ-LAI for different vegetation types are compared in 2021. (a), (b), (c), and (d) represent spring, summer, autumn, and winter, respectively. The vertical lines represent the mean value of MODIS LAI (red) and HiQ-LAI (green), respectively.

Furthermore, we conducted a histogram of the numerical distribution (Fig. 8) of the two products under different vegetation types globally in 2021 by season. A similar phenomenon can be found in Fig. 7, with the most notable discrepancies observed in the numerical distribution of type B5. Moreover, substantial differences were across various seasons for B5, and the red vertical line representing the mean of MODIS LAI is consistently positioned to the left of the green vertical line representing the mean of HiQ-LAI. The distribution of the B1-B4 biome remained highly consistent throughout different seasons with minimal variation. Conversely, biomes of B6 - B8 displayed differences in numerical distribution between spring and summer, demonstrating an opposite trend. In spring, the mean value of MODIS LAI (red vertical line) was consistently higher than that of HiQ-LAI (green vertical line), while in summer, the red vertical line was always located on the right of the green line.

## 4.4 LAI Trend Comparison

To compare the vegetation change trend on a global scale between MODIS LAI and HiQ-LAI products, we computed the LAI mean from 2000 to 2022. By determining the slope of the time series through the fitting and utilizing the Mann-Kendall (MK) test to discern the significance and monotonicity of trends, we obtained the spatial trend change map for this period (Fig. 9), revealing regions exhibiting greening trends (positive slope) or browning trends (negative slope) in vegetation. The spatial trend change map provides valuable insights into the dynamics of vegetation across the globe. Both products exhibit similar spatial patterns in greening trends, particularly in global hotspots such as China and India. Similarly, the browning trend is observed in similar locations in both products. From MODIS to HiQ-LAI, the proportion of insignificant increased from 37.03% to 39.95%. The greening trend and browning trend of the two products were 60.24% (MODIS)/ 56.69% (HiQ) and 2.73% (MODIS)/ 3.36% (HiQ), respectively. The slight difference between the two products is mainly concentrated in high-latitude areas.

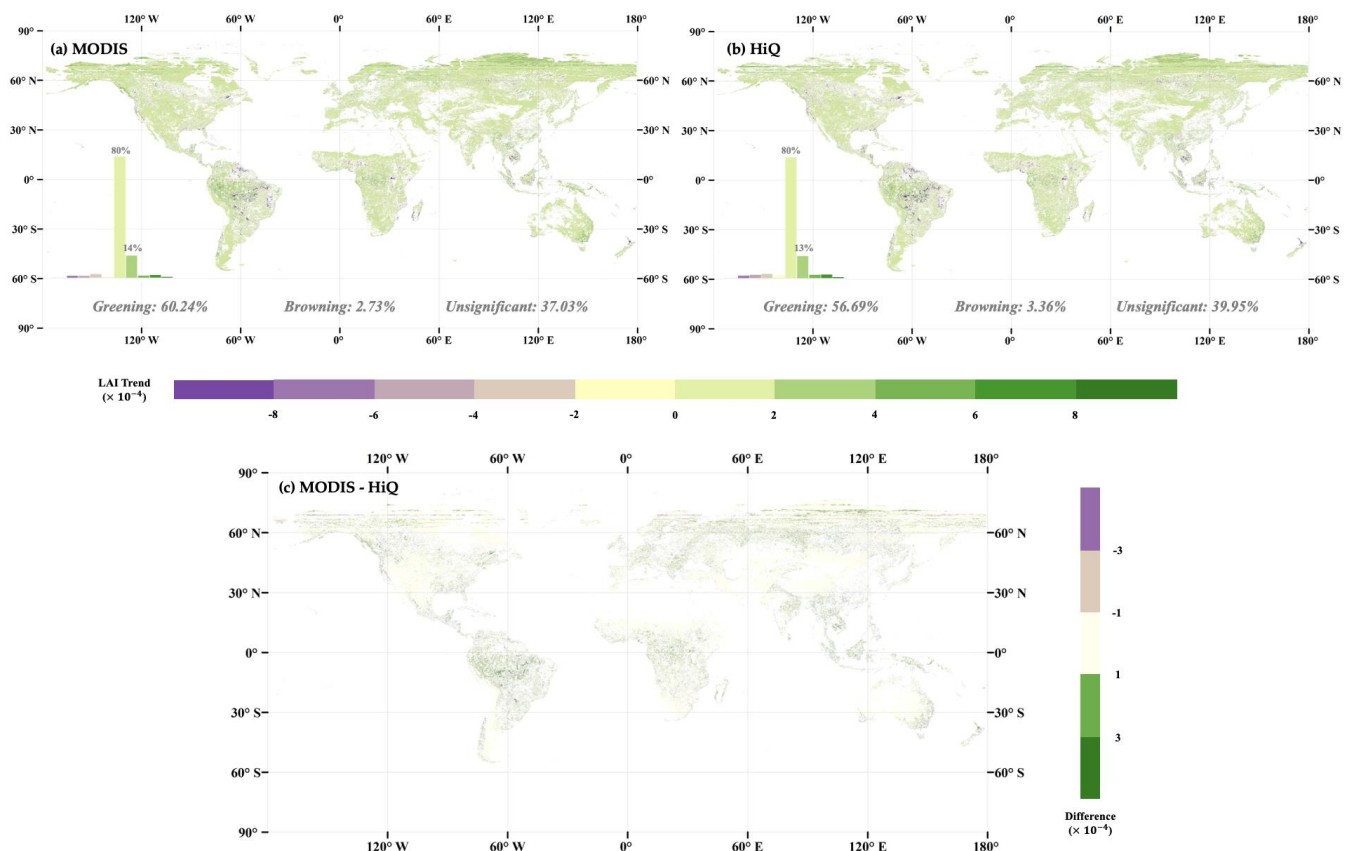

**Figure 9. Global maps of LAI trends between MODIS LAI (a) and HiQ-LAI (b) during 2000 − 2022. The Theil–Sen's slope (TS) method and Mann-Kendall (MK) test were used to calculate these results. (c) Difference of LAI trends between MODIS LAI and HiQ-LAI.**

**4.5 Improvements of Time Series Stability**

Typically, LAI exhibits continuous variations throughout the year without significant fluctuations (Zou et al., 2022). However, the annual seasonal variation curve of MODIS LAI shows pronounced abnormal fluctuations (e.g., sudden increases and decreases). This phenomenon is attributed to the independent pixel-by-pixel inversion process of MODIS LAI daily. When affected by atmospheric conditions, sensor malfunctions, and retrieval algorithm uncertainties, MODIS LAI experiences poor spatiotemporal consistency and accompanies increased noise level (Yan et al., 2021b; Fang et al., 2019). Consequently, it fails to accurately capture the long-term trend of LAI variations, thus restricting its application in crops modelling and prediction (Fang et al., 2011; Ines et al., 2013; Zhuo et al., 2022), climate change and vegetation dynamics research (Zheng et al., 2021; Chen et al., 2021), as well as long-term ecological environment monitoring and assessment (Dhorde and Patel, 2016; Mariano et al., 2018). From the ground-based direct validation results (Sect. 4.1), it is evident that HiQ-LAI effectively improves the data quality of the original MODIS LAI. It reduces apparent error of the raw LAI retrievals and generates smoother time series LAI profiles that align better with expected phenological patterns. However, the GBOV site data are limited by spatial scale and resources, making it insufficient for comparing temporal changes on a global scale. Therefore, we utilize TSS (Zou et al., 2022), a quantifiable metric for time series stability, to further understand the temporal stability performance of the two products. The HiQ-LAI encompasses abundant smaller TSS pixels (Fig. 10), particularly in regions near the equatorial characterized by EBF vegetation cover type. The proportion of low TSS globally increases from 31.8% (MODIS) to 78.8% (HiQ). This confirms that our proposed HiQ-LAI product substantially improves the stability of time series compared to the original MODIS LAI, resulting in an overall enhancement in data quality. This will provide more reliable data support for the research and applications of ecology, climatology, land-use planning, and other fields.

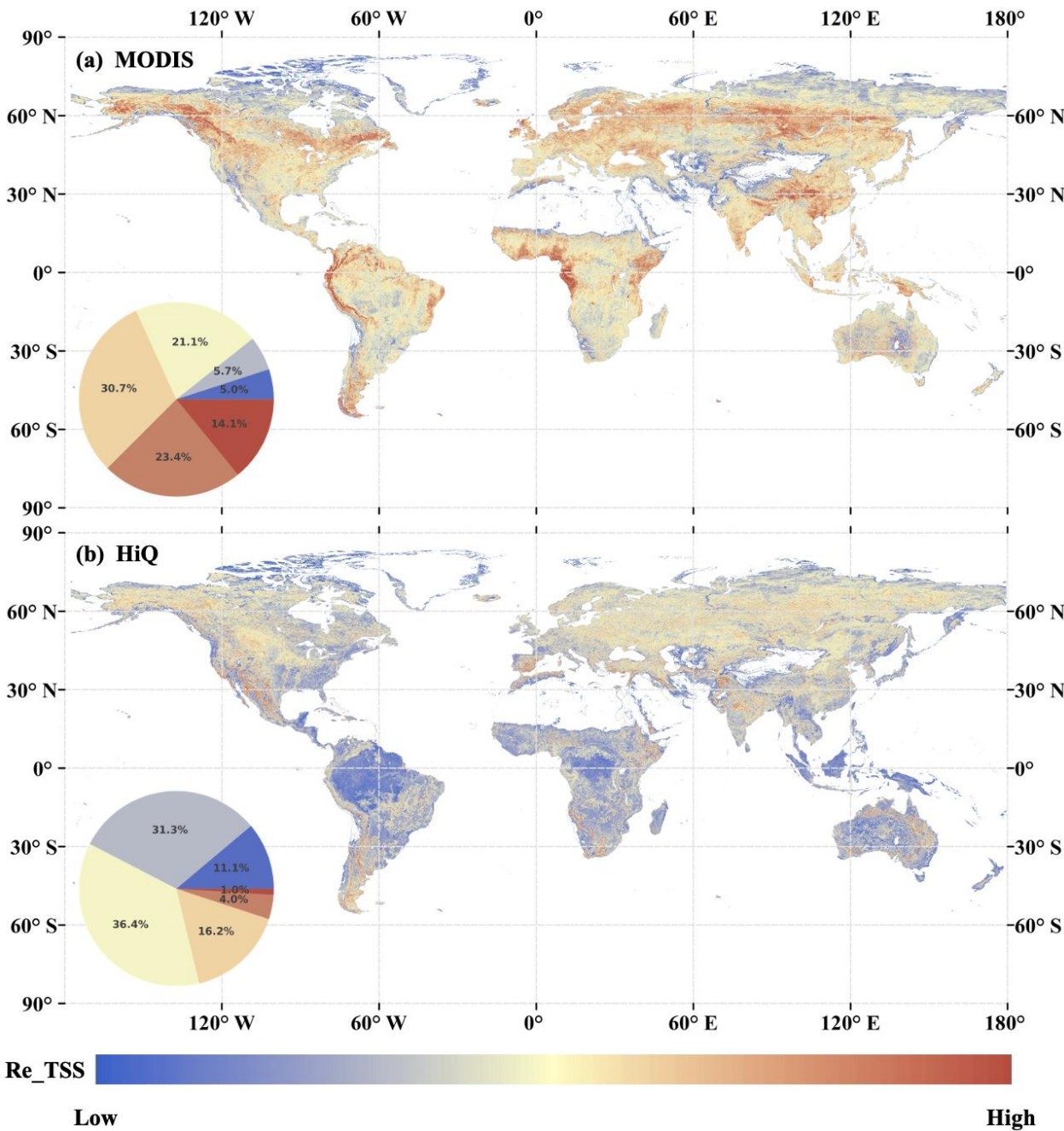

**Figure 10.** Global spatial distribution of cumulative Relative TSS (Re_TSS) for MODIS LAI (a) and HiQ-LAI (b) within 2021 (containing 46 data periods).

## 4.6 Relationship between Improvement and Raw Data Quality

Ground verification also reveals a consistent pattern between low MQA values and high LAI retrievals. Regions with high LAI values are mainly observed in tropical regions where there is greater cloud cover and aerosol load, which is prone to signal saturation (Yan et al., 2016b; Samanta et al., 2012a). In the first step of our algorithm, pixel quality is assessed based on the algorithm path and LAI standard deviation of the main algorithm, where the saturation phenomenon reduces the MQA value. Thus, this phenomenon can be attributed to the overestimation of MODIS LAI retrievals due to signal saturation (Heiskanen et al., 2012). Moreover, other studies have confirmed that the main algorithm's saturation tends to yield higher MODIS LAI (Yuan et al., 2011; Lin et al., 2023). We graded pixels according to the MQA value (Fig. 11) and observed that the improvement effect on LAI retrievals became more pronounced as the quality level decreased. In the Poor-Quality level, HiQ-LAI exhibited a 17.81% increase in $R^2$ and an 18.99% reduction in RMSE compared to MODIS LAI. However, the RMSE reduce by only 2.11% in the Good-Quality level. HiQ-LAI substantially enhances the quality of LAI retrievals affected by observation conditions and inversion algorithms while maintaining the same with pixels with good quality in the original MODIS LAI. This highlights the effectiveness of our proposed algorithm in improving the spatiotemporal consistency of LAI products.

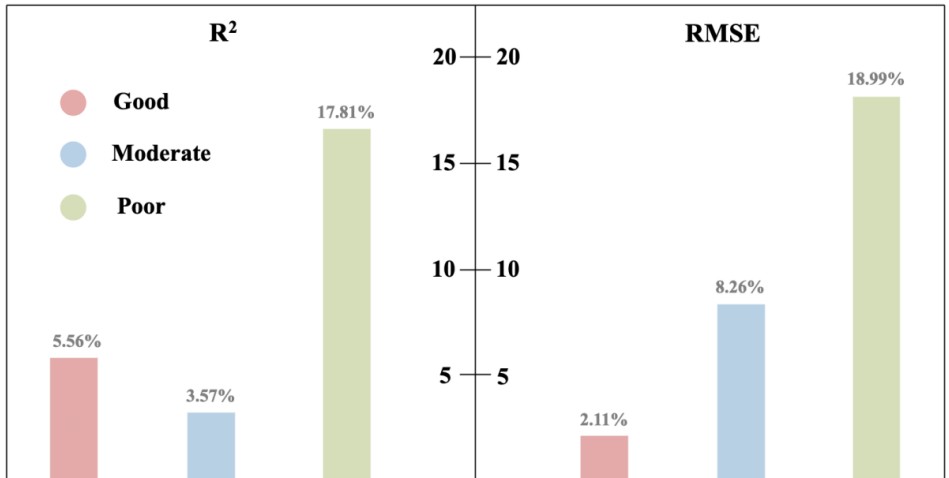

**Figure 11. Comparison of improvement percentage of RMSE and $R^2$ between MODIS and HiQ-LAI within different quality classification (good: MQA ≥ 9; moderate: 7 ≤ MQA < 9; and poor: MQA < 7) based on DIRECT 2.1 sites reference (99 sites and 268 measurements).**

## 4.7 Detailed Analysis in Equatorial Regions

Due to the long-term effects of extensive cloud cover contaminant, high aerosols, and dense vegetation saturation, the availability of high-quality observation data from optical remote sensing in tropical regions is severely limited (Huete et al., 2002; Yuan et al., 2011; Xu et al., 2018). Among these regions, the Amazon forests, in particular, have a significant impact on global climate (Cox et al., 2013; Jimenez et al., 2018; Guimberteau et al., 2017), carbon and water cycles (Marengo and

Espinoza, 2016; Poulter et al., 2014; Yang et al., 2018). MODIS LAI retrievals are calculated independently pixel-by-pixel and daily. The varying observation conditions between adjacent time windows introduce more uncertainty, leading to relatively

poor consistency in both temporal and spatial dimensions. Spatial pattern changes of MODIS LAI in short-term time series (Fig. 13) show that some higher values suddenly decrease in the next period, while some lower values suddenly increase. This phenomenon is unusual in heavily vegetated tropical regions, as previous studies have concluded that LAI in Amazon forest areas is seasonal (Hashimoto et al., 2021; Samanta et al., 2012b; Myneni et al., 2007). Correspondingly, regions with sudden LAI changes exhibit lower MQA values, indicating that our method effectively identifies retrievals with quality issues.

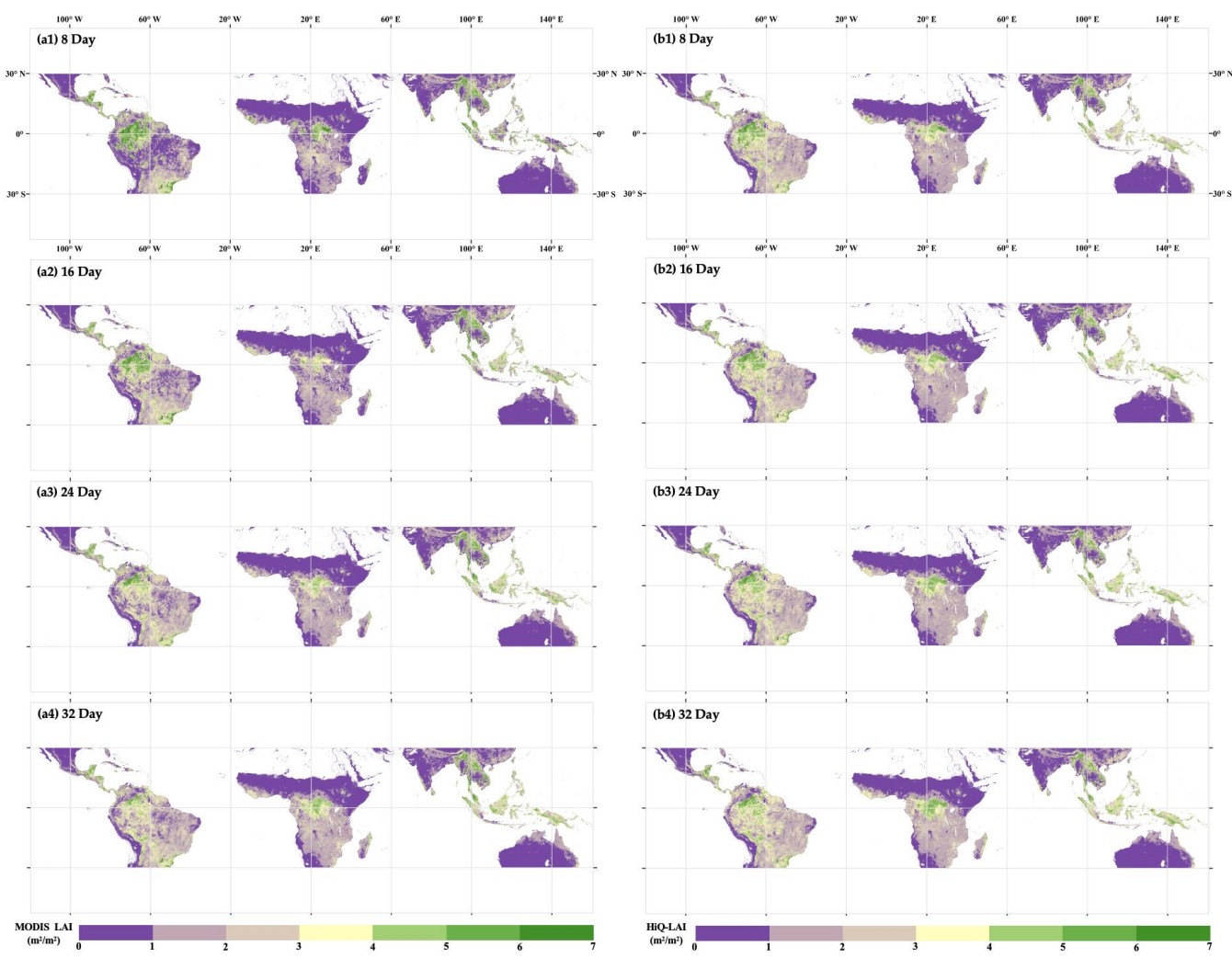


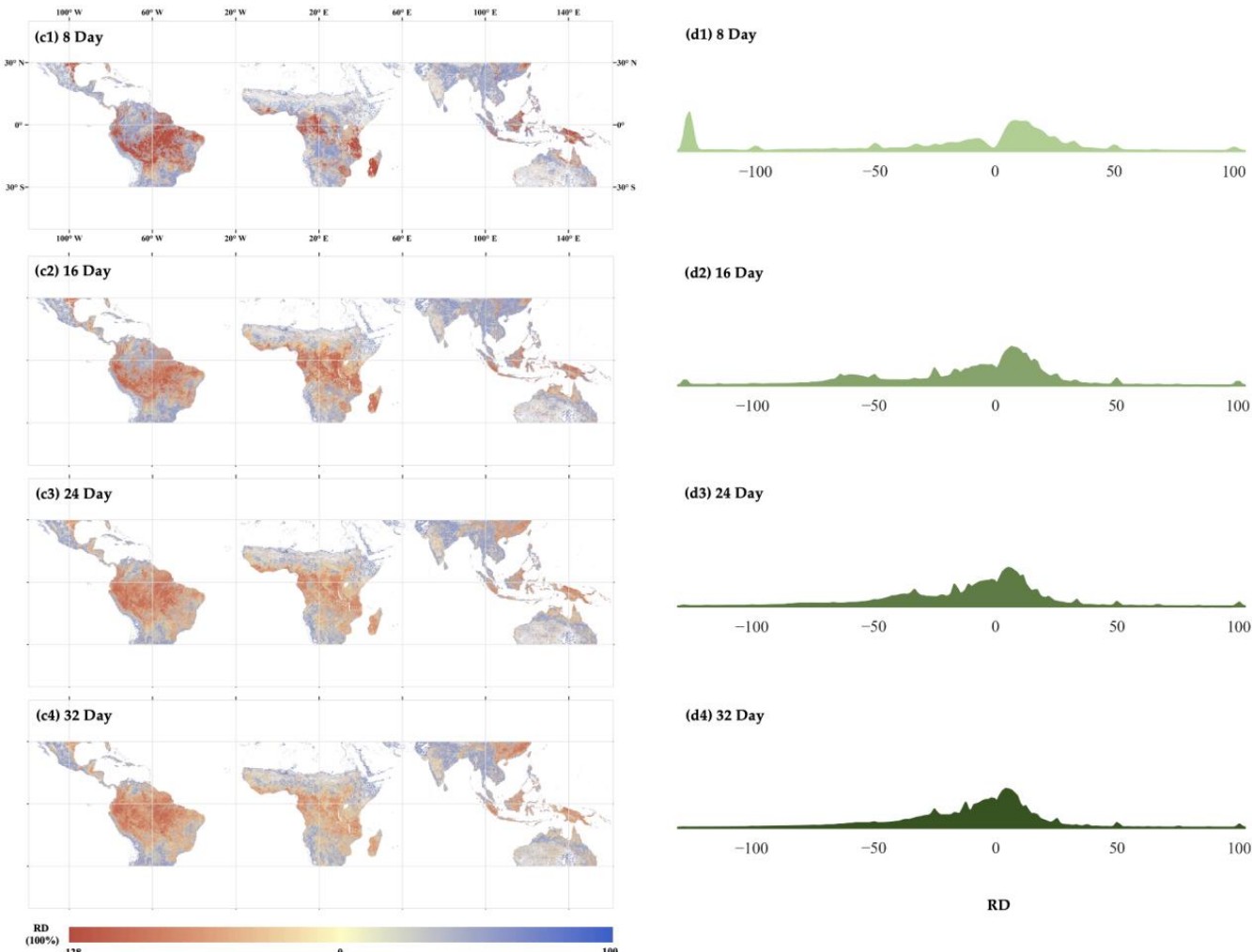

**Figure 12. Spatial distribution of MODIS LAI (a1-a4) and HiQ-LAI (b1-b4) in equatorial region within different composite day. Spatial distribution of RD (Relative Difference, c1-c4) and density distribution of RD (d1-d4) in equatorial region within different composite day. The DOY for the first 8 days of data is 2021041.**

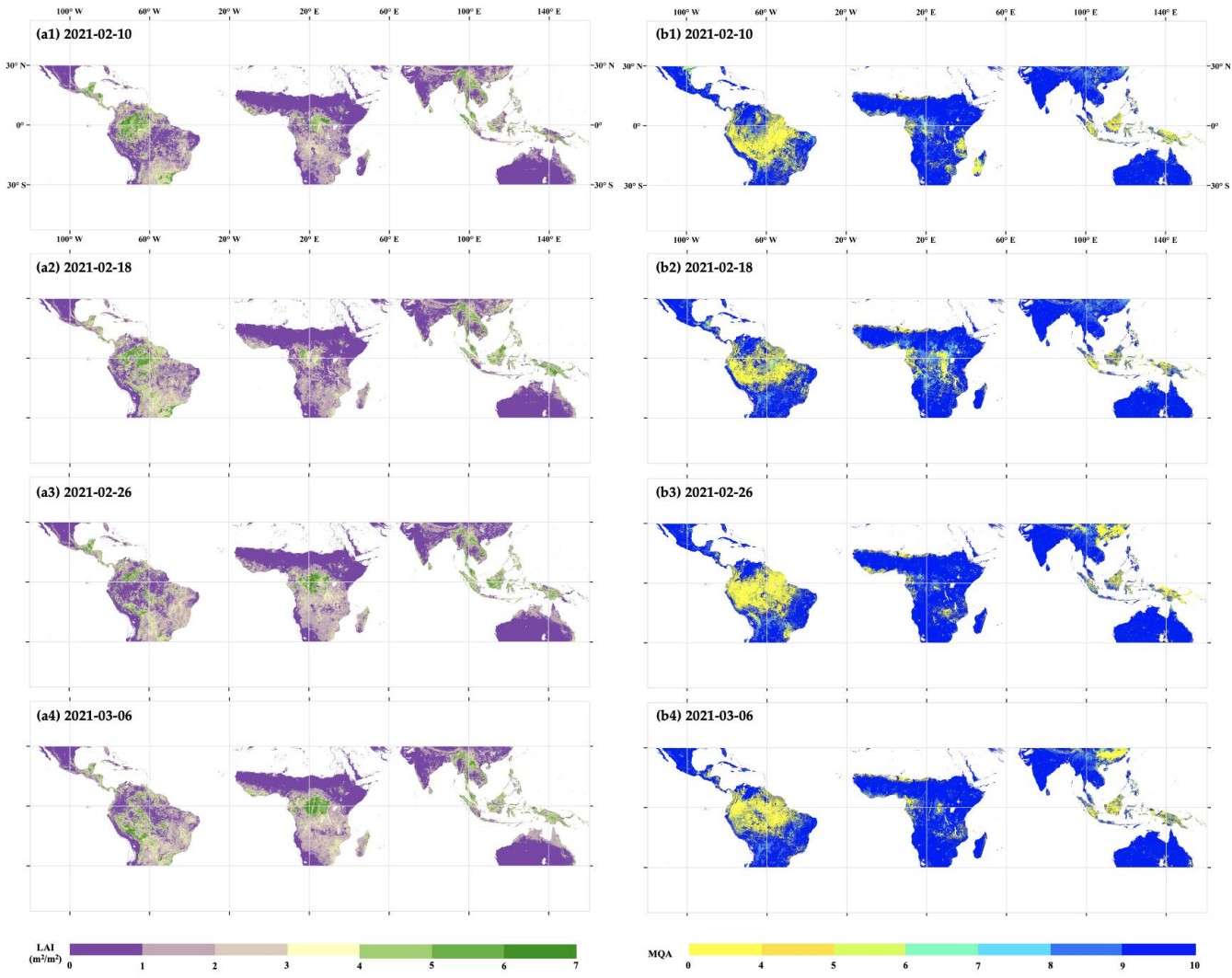


**Figure 13. Spatial distribution of MODIS LAI (a1-a4) and MQA (b1-b4) values over equatorial region.**

The differences between the two products are mainly observed in the Amazon forests regions of South America, Indonesia, and Central Africa. These areas are primarily characterized by EBF vegetation type (Fig. 1), which aligns with the differences observed in the biome comparisons (Fig. 7, Fig. 8). The spatial pattern difference between MODIS LAI (Fig. 12 (a1)) and

HiQ-LAI (Fig. 12 (b1)) at an 8-days is the most pronounced. However, the spatial distributions of MODIS LAI and HiQ-LAI gradually become consistent as the synthesis period increases (Fig. 12). The 8-day Relative Difference (RD) density distribution shows a distinct peak towards the far left. Nevertheless, the peak flattens out as the synthesis period increases, and the RD becomes more concentrated around zero. No significant spatial distribution change was observed in HiQ-LAI with the increasing synthesis period, and the 32-day synthesis data of MODIS LAI exhibited a similar spatial pattern to the 8-day

product of HiQ-LAI. This evidence indirectly indicates that compared to the original MODIS LAI, our products exhibit fewer

abnormal fluctuations in the time series and greater stability in product quality. In terms of time series stability and anomalies (Fig. 14), HiQ-LAI also outperforms MODIS LAI. By comprehensively considering pixel quality information and prior spatiotemporal correlation information, our algorithm effectively compensates for the abnormal fluctuations caused by differences in observation conditions. Through the optimization of MODIS LAI retrievals with poor quality, our algorithm

substantially improves the quality of original MODIS LAI products and enhances their applicability in tropical regions. The increased stability and reliability of HiQ-LAI product provide more reliable data support for research in tropical regions.

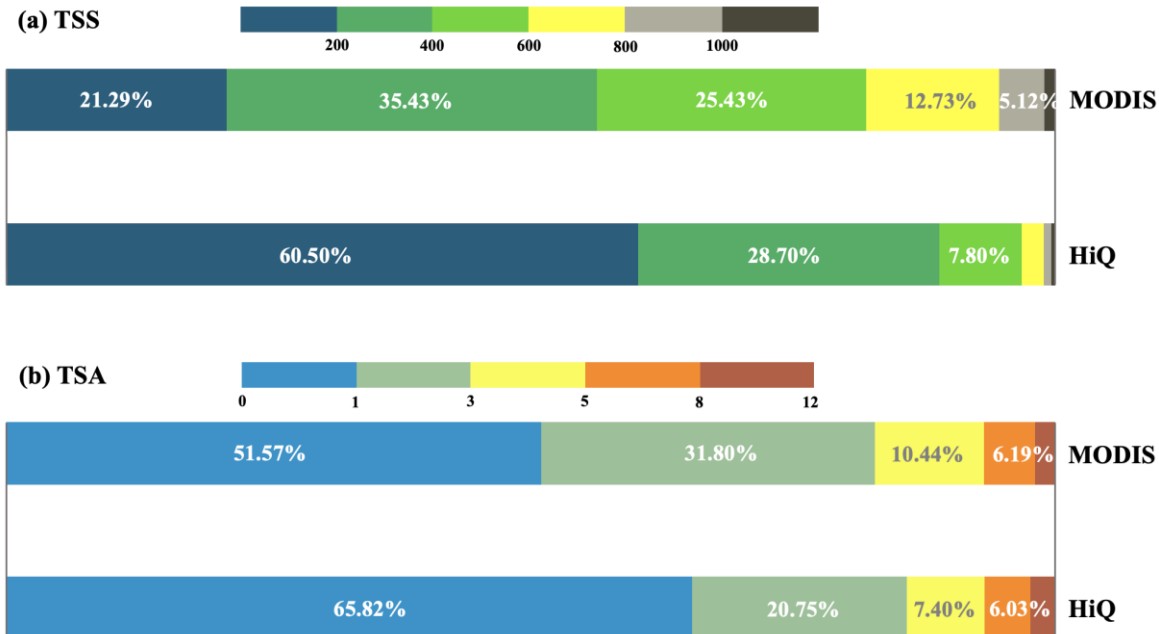

**Figure 14. The proportion of cumulative TSS (a) over 32 days and cumulative TSA (b) in 2021 between MODIS LAI and HiQ-LAI in the equatorial region.**

## 5 Potential Applications and Limitation of the HiQ-LAI


As one of the most widely used LAI products, MODIS LAI possesses irreplaceable advantages. In this study, we employed the algorithm (STICA) based on the quality information and spatiotemporal correlation to effectively identify retrievals with poor quality and improve their quality while maintaining the original physics-based (radiation transfer model, RTM) LAI generation process. The newly generated product, HiQ-LAI, is more reliable and consistent in both spatial and

temporal compared to the original MODIS LAI. Furthermore, the implementation of STICA solely relies on MODIS products without additional data requirements, mitigating the added uncertainty associated with using multiple data sources. Long-term time series of LAI play a crucial role in forest monitoring, climate model simulation (Boussetta et al., 2013; Richardson et al., 2013; Tillack et al., 2014) global water regulation, carbon and energy cycles (Sellers et al., 1997a; Chen et al., 2020). HiQ-LAI is expected to facilitate the advancement of these models by providing more accurate and reliable data sources.

Additionally, due to the algorithm's high adaptability, it can also be applied to improve the data quality of other vegetation state parameters such as FPAR, NDVI, EVI, etc.

We implemented the algorithm on the GEE cloud computing platform. Leveraging the powerful computing capabilities of the platform, users can easily conduct long-term time series and large-scale, even global-scale research and applications using this dataset. To facilitate users' understanding of the differences between the two products at pixel scale, we provide the absolute difference between the two products as a data layer for users' reference.

Despite our efforts to enhance the overall data quality through spatiotemporal correlation, uncertainties still exist. This algorithm utilizes auxiliary MODIS land cover data (MCD12Q1) to identify pixels with the same biome type. Therefore, the classification uncertainty associated with this auxiliary data can affect the algorithm's accuracy. Additionally, the limited availability of high-quality observation samples in the high latitudes of the Northern Hemisphere, constrained by solar zenith angle and environmental conditions, poses challenges. Insufficient ground observation data in these regions further hinders accurate uncertainty assessments. Another source of uncertainty arises from the calculation of MQA values. The weight assigned to each pixel is not only influenced by its spatiotemporal proximity but also closely tied to the MQA value. The more observed information provides more comprehensive information, enabling a more accurate assessment of pixel quality. Thus, future iterations of the product could incorporate additional pixel quality information, including environmental variables (e.g., cloud cover and aerosol) that substantially impact LAI.

## 6. Data availability

The High-quality Leaf Area Index (HiQ-LAI) is now accessible as an open dataset, providing two different spatial resolutions. The HiQ-LAI product with a spatial resolution of 500 meters and a temporal resolution of 8 days are hosted on the GEE platform (https://code.earthengine.google.com/?asset=projects/verselab-398313/assets/HiQ_LAI/wgs_500m_8d) for user access. Including this dataset on the GEE platform will greatly benefit the GEE community, simplifying access to and utilization of this valuable resource. Datasets with a spatial resolution of 5km and a temporal resolution of 8 days were derived by upscaling the original 500m data using the nearest-neighbour method can be found on Zenodo (https://doi.org/10.5281/zenodo.8296768) (Yan et al., 2023). The HiQ-LAI product encompasses 5-layer scientific datasets, containing LAI, original quality control information, relative-TSS of MODIS and HiQ LAI, and the absolute difference between HiQ-LAI and MODIS LAI. Detailed quality control information can be referred to the MODIS C6 LAI User's Guide (Myneni, 2020). The original values have been adjusted to integers considering the data storage size, and all layers are stored as uint8 data types. The LAI layer has a valid range from 0 to 100, where each value represents a tenth of the original value. The relative-TSS of MODIS LAI and HiQ-LAI have valid ranges spanning from 0 to 10, with each value corresponding to 0.001 units. Lastly, the LAI difference layer (LAI_Diff) applies a scale factor of -100. Moreover, we utilized bicubic interpolation on a 500m scale to create the 5km and 8days dataset, which is available on the GEE platform (https://code.earthengine.google.com/?asset=projects/verselab-398313/assets/HiQ_LAI/wgs_5km_8d_Bicubic). This method

involves considering information beyond just the nearest pixel and incorporating additional data from surrounding pixels to calculate new pixel values, which maximally preserves the pixel-level details from the original scale. Additionally, we aggregate the pixels by calculating the mean of a 5 km x 5 km window. Then, nearest-neighbour resampling can be carried out on this aggregated dataset to reproject it into the spatial resolution of 5km. The final resulting 5km dataset has been successfully stored in GEE https://code.earthengine.google.com/?asset=projects/verselab-398313/assets/HiQ_LAI/wgs_5km_8d_NearNei. More details about HiQ-LAI can be accessed at https://github.com/tiramisu18/HiQ-LAI. Users are encouraged to refer to the data description document for guidance on value restoration.

## 7. Conclusion

The Moderate Resolution Imaging Spectroradiometer (MODIS) Leaf Area Index (LAI) retrievals are calculated independently for each pixel and a specific day. However, cloud, snow, aerosol pollution, sensor failure, and uncertainties of the retrieval algorithm lead to MODIS LAI products having poor spatiotemporal consistency and accompanied by abnormally high noise. To address these limitations and improve the quality and spatiotemporal consistency of existing MODIS LAI product on a global scale, we utilize the Spatiotemporal Information Compositing Algorithm (STICA) that leverages pixel quality information, spatiotemporal correlation, and original observation information to improve the MODIS LAI retrievals with poor quality. By considering multiple dimensions of information to compensate for the deficiency of using only temporal information, High-Quality Reprocessed MODIS LAI Dataset (HiQ-LAI) achieve better spatio-temporal consistency and provide global coverage from 2000 to 2022.

Direct ground verification demonstrates that HiQ-LAI exhibits higher accuracy and is closer to the ground references compared to the original MODIS LAI. MODIS LAI tends to overestimate LAI values for pure forests, especially in areas with low Multiple Quality Assessment (MQA) values, due to signal saturation issues. When comparing the LAI time-series profile, it becomes evident that MODIS LAI is greatly affected by the observation conditions, resulting in abnormal fluctuations in the time series. In contrast, HiQ-LAI effectively mitigates these issues, generating smoother LAI time-series curves that align well with expected phenological patterns. Based on the references from the DIRECT 2.1 site, the improvement rate of LAI retrievals gradually increased with the decreasing data quality. The global trend analysis reveals a general greening trend in most vegetation areas for both products. MODIS LAI product is widely used due to its clear theoretical basis and satisfactory verification results. Our original intention is to maintain consistency with the original high-quality MODIS LAI while enhancing the accuracy of LAI retrievals with poor quality. Our results confirm that both products generally exhibit similar spatial patterns globally, but differences emerge in certain regions, particularly near the equatorial (e.g., South America, Indonesia, and the Amazon rainforest region of Central Africa). These regions are characterized by Evergreen Broadleaf Forest (EBF) and are affected by multi-cloud coverage, high aerosol concentration, and near-infrared saturation throughout the year, substantially impacting the accuracy of MODIS LAI retrievals. Changes in observation conditions between adjacent time windows introduce more uncertainty, leading to relatively poor spatiotemporal consistency of LAI. To address this challenge,

we introduced prior knowledge and leveraged original high-quality observations and spatiotemporal correlation to optimize LAI retrievals with poor quality. The generated HiQ-LAI exhibits fewer abnormal fluctuations in time series and more consistent spatial patterns in regions with obvious differences, demonstrating stronger stability and reliability of product quality. These findings indicate that our method effectively improves the spatiotemporal consistency of LAI products. The processing of HiQ-LAI is based on the GEE cloud computing platform, which provides users with easy access to long-term time series and large-scale global research and applications. Moreover, we plan to extend this algorithm to other MODIS products (e.g., FPAR, NDVI) that characterize vegetation state parameters to offer more comprehensive and accurate data support for vegetation monitoring and ecological research.

**CRediT Author Statement.**

**KY, JW:** Methodology, Conceptualization, Software, Formal analysis, Writing - Original Draft, Visualization. **KY:** Methodology, Writing - Review & Editing, Funding acquisition, Project administration, Investigation. **JW, RP KY:** Writing - Review & Editing, Formal analysis. **JP:** Investigation. **XC, LF:** Investigation, Supervision. **MW:** Supervision, Methodology, Resources. **RM:** Conceptualization, Resources, Supervision.

**Competing Interests.**

The authors declare that they have no known competing financial interests or personal relationships that could have influenced the work reported in this study.

**Disclaimer.**

Publisher's note: Copernicus Publications remains neutral with regard to jurisdictional claims in published maps and institutional affiliations.

**Acknowledgments.**

We are grateful to the editors and anonymous reviewers for their constructive comments and suggestions for improving the manuscript. We gratefully acknowledge the Google Earth Engine (https://earthengine.google.com/).

**Financial Support.**

This work was supported in part by National Natural Science Foundation of China under Grant 42271356 and the National Natural Science Foundation of China Major Program under Grant 42192580.

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
