# Peer review of "HiQ-LAI: A High-Quality Reprocessed MODIS LAI Dataset with Better Spatio-temporal Consistency from 2000 to 2022"

_Earth System Science Data, 2023_

## Author Comment (AC1)

**Response to Reviewer:**

(Retype your comments in *italic font* and then present our responses to the comments)

*This manuscript introduces the newly generated global High-Quality LAI (HiQ-LAI) Product at 500 m/5 km and 8 days resolution from 2000 to 2022. This product was generated on the GEE platform using a well-validated Spatio-Temporal Information Composition Algorithm (STICA). The HiQ-LAI can be considered as a reprocessed and value-add version of the raw official MODIS LAI products. Evaluation results demonstrate a significant improvement compared to raw MODIS LAI in terms of RMSE and Bias, enhanced temporal stability, and superior continuity especially in equatorial regions where optical remote sensing typically struggle to achieve good performance. HiQ-LAI keeps the same data format and similar quality control information with MODIS LAI, which is very convenient for data users. For me, this paper is well organized and the new product should be useful to the community of Climate Data Record (CDR). This new version of global LAI has the potential to better replace the MODIS raw product (MOD15A2) for most applications and thus desires to be published. However, there are still some minor points that need to be modified to improve the paper. Please improve these issues as follows:*

Response:Thank you so much for providing positive feedback and detailed suggestions to improve our paper. We have thoroughly revised our manuscript and addressed all the questions and concerns raised. Our responses are given below.

*1. Authors are encouraged to employ precise terminology when addressing uncertainty and accuracy in the manuscript. According to GCOS/CEOS, accuracy is defined as the proximity between the product and the reference values (doi: 10.1016/j.envsci.2015.03.018).*

Response:Thanks for your comments. We had checked our manuscript and revised.

" It reduces apparent error of the raw LAI retrievals and generates smoother time series LAI profiles that align better with expected phenological patterns."

*2. As the manuscript said, most existing filtering methods could artificially remove land surface real disturbances (e.g., forest fire, land cover change). In such cases, how does HiQ-LAI perform?*

Response:While most existing filtering methods effectively utilize temporal and QC layers information, they often overlook spatial correlation information. Consequently, although the LAI profile may appear smoother, genuine land surface LAI anomalies may be artificially removed. Therefore, we fully utilized the quality, spatiotemporal information, and relative original observation records, and these pieces of information are weighted and averaged according to our fusion strategy. More robust results are obtained by considering multiple dimensions of information to compensate for the limitations of using a single information source and by preserving as real LAI anomalies as possible.

*3. In the abstract "However, MODIS LAI retrievals are calculated independently for each pixel and a specific day, resulting in high noise levels in the time series and limiting its applications." Can the expression here be more precise, for example, in the regions of XXX.*

Response: Thanks for your suggestions, and we had overwritten this sentence.

"However, MODIS LAI retrievals are calculated independently for each pixel and a specific day, resulting in high noise levels in the time series and limiting its applications in the regions of optical remote sensing is severely limited."

4. Page 2, Lines 50-55: "The long time series MODIS LAI dataset has made significant contributions to landmark studies on "Greening the Earth" phenomena", suggest replace 'long time series' to 'long-time series', 'on "Greening the Earth" phenomena' to 'on the "Greening the Earth" phenomena'.

Response:Thanks for your suggestion. We have overwritten this sentence.

"The long-time series MODIS LAI dataset has made significant contributions to landmark studies on the "Greening the Earth" phenomena, the possible causes of large-scale vegetation dynamics, and the relationship between vegetation dynamics and global climate change or human activities."

5. Page 4, Figure 1: suggest adding a schematic representation of the study area's location of Section 5.3 on this global map.

Response:Thank you for your comments and have been modified the Figure 1 according to your suggestions.

[Figure]

**Figure 1. Geographical distribution of the selected sites. The background color indicates the biome types of the 2021 MCD12Q1 classification scheme. The red hexagon, yellow triangle, pink dots, and red frame represent the GBOV sites, DIRECT 2.1 sites, BELMANIP 2.1 sites, and Equatorial Regions, respectively.**

6. Page 7, Lines 190-195: "we utilized the GBOV LAI measurements from a total of 29 sites spanning from 2013 to 2021 as our ground reference LAI.", add reference here.

Response:Thanks for your comments and we added the references as suggested.

"In this study, we utilized the GBOV LAI measurements from a total of 29 sites spanning from 2013 to 2021 as our ground reference LAI (Bai et al., 2019; Brown et al., 2020)."

7. Page 7, Lines 195-205: "Furthermore, we compared MODIS LAI and HiQ-LAI in 2021 using the BELMANIP V2.1 sites (445 in total).", "Additionally, we used DIRECT V2.1 ground measurements in this research." Please add reference here too.

Response:Thanks for your comments and we added the references as suggested.

"Furthermore, we compared MODIS LAI and HiQ-LAI in 2021 using the BELMANIP V2.1 sites (445 in

total) (Baret et al., 2006)."

"Additionally, we used DIRECT V2.1 ground measurements in this research (Morisette et al., 2006; Garrigues et al., 2008)."

8. Page11, Table 1: the decimal places are not consistent, suggest changing them and unifying them. Besides, if there is insufficient data to compute RMSE and R2 for a particular site, it is recommended to fill the table with '—' rather than '0.00'.

Response:Thanks for your suggestions, we had unified the decimal places and used '---' instead of '0.00'. Besides, we added information about the biome type of site, the Relative RMSE (RRMSE), and Bias in Table 1.

**Table 1.** Comparison of MODIS LAI and HiQ-LAI over GBOV sites

| Biome Type | Site | M_RMSE | H_RMSE | M_R$^2$ | H_R$^2$ | M_RRMSE (%) | H_RRMSE (%) | M_Bias | H_Bias |
|---|---|---|---|---|---|---|---|---|---|
| Grasslands | CPER | 0.44 | 0.40 | 0.21 | 0.22 | 75.90 | 69.32 | -0.39 | -0.37 |
| | KONA | 1.26 | 1.19 | 0.47 | 0.68 | 102.32 | 96.31 | -1.20 | -1.16 |
| | MOAB | --- | --- | --- | --- | --- | --- | --- | --- |
| | ONAQ | 0.31 | 0.29 | 0.06 | 0.04 | 95.57 | 90.81 | -0.20 | -0.19 |
| | SRER | 0.21 | 0.21 | 0.84 | 0.86 | 45.16 | 44.94 | -0.11 | -0.05 |
| | STER | --- | --- | --- | --- | --- | --- | --- | --- |
| | WOOD | 1.29 | 1.19 | 0.55 | 0.66 | 110.19 | 102.07 | -1.24 | -1.15 |
| Forests | HARV | 0.82 | 0.53 | 0.72 | 0.83 | 19.37 | 12.41 | -0.24 | -0.15 |
| | TALL | 1.29 | 1.14 | 0.77 | 0.81 | 38.72 | 34.10 | -1.08 | -0.96 |
| | TUMB | 1.33 | 1.31 | 0.94 | 0.95 | 77.58 | 76.23 | -1.19 | -1.19 |
| Grasses & Shrubs | JORN | 0.06 | 0.16 | 1.00 | 1.00 | 14.95 | 35.91 | -0.03 | 0.01 |
| | VASN | --- | --- | --- | --- | --- | --- | --- | --- |
| Crops & Savannas | BLAN | 0.53 | 0.34 | 0.41 | 0.71 | 36.72 | 23.76 | -0.44 | -0.31 |
| | LAJA | 0.32 | 0.17 | 0.99 | 0.20 | 25.31 | 13.42 | -0.24 | -0.13 |
| Grasses & Savannas | GUAN | 0.72 | 0.49 | 0.36 | 0.43 | 22.23 | 15.13 | -0.59 | -0.33 |
| | JERC | 0.72 | 0.80 | 0.70 | 0.81 | 24.69 | 27.42 | 0.61 | 0.76 |
| | LITC | 0.85 | 0.68 | 0.58 | 0.66 | 149.71 | 120.21 | -0.83 | -0.66 |
| | NIWO | 0.49 | 0.31 | 0.68 | 0.64 | 66.89 | 42.19 | -0.47 | -0.30 |
| Savannas & Forests | BART | 0.75 | 0.46 | 0.80 | 0.89 | 19.60 | 11.87 | -0.14 | 0.06 |
| | DELA | 1.16 | 1.10 | 0.05 | 0.03 | 26.67 | 25.17 | 0.42 | 0.91 |

| | | | | | | | | |
|---|---|---|---|---|---|---|---|---|
| DSNY | 0.76 | 0.89 | 0.75 | 0.86 | 30.34 | 35.54 | 0.75 | 0.89 |
| HAIN | 1.11 | 0.45 | 0.50 | 0.88 | 24.83 | 10.16 | 0.16 | 0.21 |
| ORNL | 0.73 | 0.68 | 0.44 | 0.55 | 18.50 | 17.05 | -0.02 | 0.39 |
| OSBS | 0.45 | 0.42 | 0.85 | 0.74 | 18.44 | 17.24 | 0.20 | 0.30 |
| SCBI | 1.05 | 0.81 | 0.52 | 0.78 | 23.00 | 17.67 | 0.41 | 0.54 |
| SERC | 0.84 | 1.38 | 0.83 | 0.80 | 18.85 | 30.92 | 0.75 | 1.31 |
| STEI | 0.75 | 0.65 | 0.54 | 0.67 | 18.17 | 15.81 | -0.20 | 0.42 |
| UNDE | 0.42 | 0.28 | 0.03 | 0.47 | 9.50 | 6.40 | 0.07 | 0.14 |
| WOMB | --- | --- | --- | --- | --- | --- | --- | --- |

9. Page13, Lines 260-265: "the R2 for other pure vegetation types exceeds 0.88, and B1 and B3 surpassed 0.95. The consistency of mixed pixels is also relatively high, as indicated by an RMSE of 0.42 and an R2 of 0.86. However, B5 exhibits a significant disparity, with an R2 value of 0.15." please modify these 'R2' to 'R²'.

Response:Thank you for pointing this out and we apologize for this point. We checked all mathematical symbols and corrected the incorrect ones.

"The result demonstrates that, except for B5, the $R^2$ for other pure vegetation types exceeds 0.88, and B1 and B3 surpassed 0.95. The consistency of mixed pixels is also relatively high, as indicated by an RMSE of 0.42 and an $R^2$ of 0.86. However, B5 exhibits a significant disparity, with an $R^2$ value of 0.15."

"In the Poor-Quality level, HiQ-LAI exhibited a 17.81% increase in $R^2$ and an 18.99% reduction in RMSE compared to MODIS LAI."

10. Page15, Lines 285-290: the usage procedures for Theil–Sen's slope (TS) method and the Mann-Kendall (MK) test are not sufficiently clear, please provide a more detailed description and relevant mathematical formulas for Theil–Sen's slope (TS) method and the Mann-Kendall (MK) test in Section of the methodology.

Response: Considering the Reviewer's suggestion, we added the detailed description and relevant mathematical formulas for Theil–Sen's slope (TS) method and the Mann-Kendall (MK) test in Section 3.1.

"In this study, the Theil-Sen's slope (TS) method and Mann-Kendall (MK) test (Suhartati, 2013; Theil, 1992) were employed to extract LAI trends from the two products. The TS method computes pairwise slopes across the study period, with the median slope representing the sign and magnitude of the long-term trend. Unlike ordinary least-square linear regression, the TS trend is less susceptible to the influence of outliers. Meanwhile, the MK test is utilized to determine the significance of the trend (Kendall, 1948). The combination of TS and MK forms a robust approach for identifying trends in long-term sequential data. TS and MK are calculated as follows:

$$TS = median\left(\frac{X_j - X_i}{j - i}\right), \quad 2000 \leq i < j \leq 2022 \tag{1}$$

[revised manuscript text omitted]

---

## Author Comment (AC2)

**Response to Reviewer:**

(Retype your comments in *italic font* and then present our responses to the comments)

*The submitted manuscript presents HiQ-LAI, a reprocessed leaf area index (LAI) dataset based on the Moderate Resolution Imaging Spectroradiometer (MODIS) LAI product that attempts to improve spatiotemporal consistency. There is a need to reduce spatiotemporal noise in global LAI products, and the HiQ-LAI dataset appears to offer improvements in this respect, so there is obvious potential for it to be useful to the community. I was able to access the 5 km dataset on Zenodo (though files appeared to be in .tif format, rather than HDF as claimed on L449), as well as the 500 m collection in Google Earth Engine without issue. Although the dataset appears useful, I do have several concerns and requests for clarification.*

Response: Special thanks for your positive comments and very detailed suggestions to make the paper better. HDF format data has not been stored in GEE at present, and corresponding modifications have been made in the paper. Following the Reviewers' comments, we have carefully revised our manuscript, and adequately addressed all the questions and concerns that the referees have raised. Hope this revised manuscript has solved all your concerns.

*Main concerns:*

*1. Too little detail is provided about the Spatio-Temporal Information Compositing Algorithm (STICA) used to derive the dataset. Whilst I appreciate that STICA is described in a previous publication (https://doi.org/10.1109/TGRS.2023.3264280), a brief overview of how it actually works is still needed. The current description is very superficial, outlining only the names of each step, rather than how each step is actually achieved. For example, it is not clear what the Multiple Quality Assessment (MQA) information is. Does it relate to the 'FparLai_QC' and 'FparExtra_QC' quality indicators? Likewise, how is the MQA information actually used to weight different observations in the 'fusion strategy'?*

Response: Thanks for your suggestions, and we added a detailed description of the STICA in Sect 3.1: "Satellite remote sensing observations are often subject to uncertainties arising from climatic factors, sensor malfunctions, and other sources, resulting in varying levels of uncertainty for individual pixels. To address this issue, this approach employed multiple indicators to evaluate the uncertainty for each pixel (referred to as MQA hereafter). These indicators encompass the algorithm path, STD LAI, and Relative Time-series Stability (TSS). The algorithm path (AP) is a crucial quality index, distinguishing between the main and backup algorithms. The main algorithm offers superior quality and precision retrieval, and the weight ratio of the main algorithm and backup algorithm is determined as 6:4 in the previous study (Wang et al., 2023). STD LAI reflects the retrieval uncertainty. The AP and STD LAI are derived from the FparLai_QC and LaiStdDev layers of the original MODIS data. The third indicator, Relative TSS (RE-TSS), indicates the fluctuation of a time series (Zou et al., 2022). Following the principle of assigning a higher weight to smaller values, STD LAI and RE-TSS are incorporated into the retrieval with the main algorithm, resulting in the generation of a new quality classification indicator, MQA. Subsequently, the Inverse Distance Weighting (IDW) method is utilized on the spatial scale to calculate the weighted average of all eligible pixels (belonging to the same land cover type) within a certain spatial range of the target pixel. In this algorithm, the contribution of a pixel is determined not only by its spatial distance but also by its MQA value. In a word, pixels with closer proximity and higher MQA value make a more significant contribution to the target pixel. On the temporal scale, the Simple Exponential Smoothing (SES) method is employed to calculate the weighted average of all eligible pixels within a specific period. Pixels that are closer in time to the

target pixel and possess higher MQA values are assigned greater weights. Utilizing spatial/temporal correlation is based on spatial and temporal autocorrelation, i.e., everything is related to everything else, but near things are more related than distant things. The final step of the algorithm is to take a weighted average of the original MODIS LAI and the LAI calculated using spatial/temporal correlation, with their respective weights quantified using an indicator (TSS) that represents the temporal fluctuation of the time series (Zou et al., 2022). All processes of the method are implemented using the GEE cloud computing platform. The reprocessed LAI dataset, namely the HiQ-LAI product, has been generated with the help of the powerful cloud computing capability of GEE, covering the period from 2000 to 2022."

*2. Related to point 1, the authors should make clear what the difference is between the validation results they are presenting here and the validation results from the previous publication (https://doi.org/10.1109/TGRS.2023.3264280), which seem quite similar (e.g. Figure 6 of the previous publication vs. Figure 2 in this paper). How is the present study building on the previous work? Was the previous study limited in time and space, whereas this one is more comprehensive? It needs to be made clear to the reader.*

Response: Considering the Reviewer's suggestion, we added a detailed description of the difference between the new results presented here and the validation results from the previous publication.

"In this study, we utilized the GBOV LAI measurements from a total of 29 sites spanning from 2013 to 2021 as our ground reference LAI (Bai et al., 2019; Brown et al., 2020). A 3 km × 3 km square centered on the site location was selected as the study area (Fig. 1) so that the corresponding LAI product of each site was 36 (6 × 6) pixels. To enhance the credibility of the ground truth LAI, we filtered the ground LAI reference of these 29 sites based on the criterion that the "effective pixel" exceeded 90% and the input and output of land product value in the data aggregation process were within the specified range. This filtering process yielded a total of 818 reliable verification data points. Contrary to previous studies (Wang et al., 2023) that utilized only 2018 data from the GBOV site as a reference, this study expanded the timeline from 2013 to 2021, increased the number of sites from 24 to 29, and raised the criterion for effective pixels from 80% to 90%. These modifications were aimed at enhancing the reliability of the ground LAI data. Additionally, previous research focused on proposing and testing algorithms mainly at the tile scale, but this study migrated the algorithm to GEE for generating global long-term data series. Furthermore, the scope of analysis was also broadened to a global spatial scale and long-term time series."

*3. Clarification is required regarding the choice of nearest neighbour resampling to go from 500 m to 5 km, which appears unsuitable. Surely some form of aggregation (e.g. mean value downsampling) needs to be carried out first? Otherwise, a 5 km pixel is being assigned the value of the single 500 m pixel that is closest to its centroid. That single 500 m pixel could be surrounded by pixels with very different LAI values, unless you are in an environment that is homogeneous at 5 km (which I would suggest is unheard of in terrestrial landscapes). This could be a major source of error, and I would strongly suggest the authors revisit this aspect. A proper aggregation should be easy to implement in Google Earth Engine using the 'reduceResolution' method (https://developers.google.com/earth-engine/guides/resample).*

Response: Yep, we carefully considered the Reviewer's suggestion, and we generated the 5km data using bicubic interpolation on the Google Earth Engine (GEE). This process involves considering information beyond just the nearest pixel and incorporating additional data from surrounding pixels to calculate new pixel values. The resulting 5km dataset has been successfully stored in GEE (includes

one LAI layer) and is accessible via the following link
https://code.earthengine.google.com/?asset=projects/verselab-398313/assets/HiQ_LAI/wgs_5km_8d_Bicubic.

*4. The term 'uncertainty' is used too loosely throughout the manuscript. For example, it's stated on L177-178 that 'the approach employed multiple indicators to evaluate the uncertainty for each pixel (referred to as MQA hereafter)'. If the MQA information is indeed based on categorical quality indicators, then it does not represent 'uncertainty'. According to the International Standards Organisation's Guide to the Expression of Uncertainty in Measurement, uncertainty 'characterizes the dispersion of the values that could reasonably be attributed to the measurand' (https://www.iso.org/sites/JCGM/GUM/JCGM100/C045315e-html/C045315e.html?csnumber=50461). Later in the manuscript, it's suggested that according to validation with ground data, the HiQ-LAI dataset 'reduces uncertainties of the raw LAI retrievals' (L310). Again, this is improper use of the term 'uncertainty' – here the appropriate term is 'error' or 'apparent error' (which represents the difference between the reference value and the retrieval, see https://doi.org/10.1016/j.envsci.2015.03.018 for more details).*

Response: Thanks for your comments and we had overwritten this sentence as follows:

"It reduces apparent error of the raw LAI retrievals and generates smoother time series LAI profiles that align better with expected phenological patterns."

*5. In the comparison with GBOV data, it's stated that 'GBOV LAI measurements' are used, but it's unclear whether a correction for woody material was undertaken to go from PAI to LAI. Note that the values provided by GBOV represent PAI, not LAI (https://doi.org/10.1016/j.isprsjprs.2021.02.020). If no correction was carried out, the 'GBOV LAI' should really be relabelled 'GBOV PAI', and this difference should then be considered in the discussion of the results. Related to the GBOV data, it's also not clear whether upscaled maps or individual in situ reference measurements were used – this should be clarified.*

Response: Yep, we carefully read the 《Ground-Based Observations for Validation (GBOV) of Copernicus Global Land Products -Algorithm Theoretical Basis Document - Vegetation Products LP3 (LAI), LP4 (FAPAR) and LP5 (FCOVER)》, which mentions that "For simplicity, the terms LAI and LAIe are used interchangeably with PAI and PAIe when referring to LP3". We have added the relevant instructions in Sect 5: "Note that since in-situ measurements may be sensitive to all elements of the canopy, the resulting estimate should technically be called the term plant area index (Brown et al., 2021). Insufficient ground observation data in these regions further hinders accurate uncertainty assessments."

In addition, we added a description of the use of the data in Sect 2.3.1"The Copernicus Ground-Based Observations for Validation (GBOV) service, which is part of the Copernicus Global Land Service (CGLS), is dedicated to the development and distribution of robust in situ datasets from various ground monitoring sites for the systematic and quantitative validation of land products (Bai et al., 2019; Brown et al., 2020). A comprehensive GBOV reference measurement database has been established through quality control and raw measurements reprocessing obtained from existing in situ sites. This database includes canopy reflectance, surface albedo, LAI, FPAR, cover area, 5 cm soil moisture, and surface temperature. Currently, 29 available sites provide LAI references from 2013 to 2022. The data from this database are freely accessible to the scientific community through the GBOV portal (https://gbov.acri.fr). In this study, we used the Land Products 3- leaf area index (LP3) as reference LAI."

*6. The current ground-based validation is rather simple, and more information on the improvement of the HiQ-LAI dataset could be provided by reporting the validation statistics over all sites, but split by LAI magnitude, season, and biome, so we could clearly see when and where the improvements are. This would help back up the existing discussion.*

Response: Thanks for your comments and considering the Reviewer's suggestion, we added two figures (Figure 3 and Figure 4) to explore the accuracy comparison between two products and GBOV LAI under different vegetation types, Scatter plot distribution comparison MODIS LAI (green) and HiQ-LAI (black) with GBOV LAI reference under different seasons in this study.

Comparing the LAI difference distribution among various vegetation types (Fig. 3) revealed that HiQ-LAI exhibited a tighter concentration around the zero value, resulting in decreased RMSE across most categories, except for the third biome type (Grasses & Shrubs). Mixed savannas and forests emerged as the vegetation types with the widest MODIS LAI difference range. The enhanced HiQ-LAI notably narrowed this distribution range, although the median and mean deviated further from zero. Notably, the two biome types exhibiting the most conspicuous changes in difference distribution were pure forest and mixed crops and savannas. The verification analysis (Fig. 4), comparing both products against GBOV LAI references across different seasons, demonstrated that HiQ-LAI had superior performance over MODIS LAI throughout all four seasons and exhibited outperformance with the ground references. Analyzing the LAI density distribution revealed that MODIS LAI (green) skewed towards higher values on the right side compared to HiQ-LAI (black). This indicated that MODIS LAI predominantly occupied high-value areas. Furthermore, the RMSE and RRMSE of HiQ-LAI are always smaller than that of MODIS LAI.

[Figure]

**Figure 3. Accuracy comparison between two products and GBOV LAI under different vegetation types. The numbers at the top represent the RMSE between the two products and the GBOV LAI reference, respectively.**

[Figure]

**Figure 4. Scatter plot distribution comparison MODIS LAI (green) and HiQ-LAI (black) with GBOV LAI reference under different seasons.**

*7. In the introduction, the need for the HiQ-LAI dataset could be presented more clearly. Whilst I agree that the MODIS LAI product can be subject to noise, the claim needs to be backed up with specific examples and references to previous studies that have demonstrated this (for example https://doi.org/10.3390/rs12061017 and https://doi.org/10.1016/j.rse.2020.111935 demonstrate noisier temporal profiles from MODIS LAI when compared to the Copernicus Global Land Service LAI product from PROBA-V). Likewise, it's suggested that the noise in the MODIS LAI product leads to 'limitations on its practical applications' – again some specific examples and references are needed here.*

Response: Thanks for your suggestions, and we added the references and specific examples as suggested.

"Specifically, atmospheric conditions (e.g., cloud cover, snow, and aerosol pollution), sensor malfunctions, and the inherent uncertainties of the retrieval algorithm all introduce challenges, resulting in poor spatiotemporal consistency and high noise in MODIS LAI products (Brown et al., 2020; Fuster et al., 2020; Yan et al., 2021). Consequently, inconsistency and excessive noise impose limitations on its practical applications in research involving yield estimation, crop-growth monitoring, terrestrial carbon monitoring, and global ecosystem dynamic simulation (Li et al., 2017;

Xiao et al., 2009; Chen et al., 2020)."

*8. It's stated several times that 'MODIS LAI retrievals are calculated independently for each pixel and daily' (e.g. L67-068, L341-342), which may be true on an internal basis. However, I understood that the C6.1 MODIS LAI products available to external users (i.e. MOD15A2H, MYD15A2H and MCD15A2H) are only available as 4-day and 8-day composites. So, some clarification (and careful wording) is needed here to avoid confusion.*

Response: Thank you for pointing this out, and we reorganized this sentence.

"The best retrievals are then selected using the temporal compositing method, and the 4-day or 8-day product is generated from the daily retrievals. Therefore, MODIS LAI retrievals are calculated independently for each pixel and daily. Differences in adjacent observation conditions lead to significant uncertainty in the LAI time series."

*9. The distinction between the results and discussion seemed somewhat arbitrary, since new results were shown in the discussion. In this case, a combined results & discussion section might make more sense. It was also a bit strange to have sections starting with figures.*

Response: Thanks for your suggestions, we combined the Results and Discussion section and adjusted the order of paragraphs and figures.

*Other comments:*

*Abstract*

*L19: 'reliable validation results' -> 'extensive validation results'?*

Response: Thanks for your suggestion. We have replaced the 'reliable validation results' with 'extensive validation results'.

*L35: It would be better to avoid the term 'significantly' unless statistical significance has been tested for – 'substantially' would be more appropriate. There are similar occurrences throughout the manuscript.*

Response: Thank you for pointing this out. We checked all the 'significantly' in this paper and corrected the incorrect ones with 'substantially.'

*1. Introduction*

*L48: The meaning of 'irreplaceable characteristics' is unclear.*

Response: Thanks for your comments, and we reorganized this sentence as follows:

"Among the various time-series LAI products with global coverage, Moderate Resolution Imaging Spectroradiometer (MODIS) LAI product was among the most extensively utilized LAI datasets."

*L64 and L66: What is the 'specified thershold'?*

Response: The yellow ellipse is the uncertainty range of the $\chi^2$ distribution determined according to the error level set in the red and near-infrared bands, and the average of all the inversion values corresponding to the ellipse is taken as the output (rather than only the red dot in the **Figure** as the output). When the uncertainty of the input BRFs falls within a point on the red-NIR plane and an area, all canopy or soil patterns are considered as the candidate solutions, and the mean LAI values of these solutions are used as the output values of the main algorithm.

In order to avoid the reader's misunderstanding, we have revised this sentence:

"When the uncertainty of the input BRFs falls within a point on the red-NIR plane and an area, all canopy or soil patterns are considered as the candidate solutions, and the mean LAI values of these solutions are used as the output values of the main algorithm."

[Figure]

**Figure. Schematic illustration of the main algorithm** (Myneni, 2020). **Panel A: Distribution of vegetated pixels with respect to their reflectances at red and near-infrared (NIR) spectral bands from Terra MODIS tile h12v04. A point on the red-NIR plane and an area about it (yellow ellipse defined by a $\chi^2$ distribution) are treated as the measured BRF at a given sun- sensor geometry and its uncertainty. Each combination of canopy/soil parameters and corresponding FPAR values for which modeled reflectances belong to the ellipse is an acceptable solution. Panel B: Density distribution function of acceptable solutions. Shown is solution density distribution function of LAI for five different pixels. The mean LAI and its dispersion (STD LAI) are taken as the LAI retrieval and its uncertainty. This technique is used to estimate mean FPAR and its dispersions (STD FPAR).**

*L98: 'data value-added' -> 'value-added data' (here and throughout)?*

Response: Thanks for your comments, and we have replaced all phrases of 'data value-added' in this paper with 'value-added data'.

*L102: A reference for Google Earth Engine might be needed here (https://doi.org/10.1016/j.rse.2017.06.031).*

Response: Thanks for your comments and we added the references as suggested.

"We implemented the entire algorithm process using the Google Earth Engine (GEE) cloud computing platform (Gorelick et al., 2017) to reprocess MODIS C6.1 LAI"

*2. Materials*

*L145: Perhaps the paragraph on BELMANIP V2.1 belongs in its own section, since it's related to product intercomparison rather than validation with ground reference data.*

Response: Thanks for your suggestion. We reorganized the chapter structure to include BELMANIP V2.1 as a separate Section 2.4.

*L162: The CEOS WGCV LPV LAI good practice document should be referenced here (https://doi.org/10.5067/doc/ceoswgcv/lpv/lai.002).*

Response: Thanks for your comments and we added the references as suggested.

"By the CEOS WGCV LPV LAI good practices (Fernandes et al., 2014), the ground data were upscaled using an empirical "transfer function" between high spatial resolution radiation data and biophysical

measurements to appropriately account for the spatial heterogeneity of the site."

**3. Methods**

*L185-186: Rather than the quote, it might be clearer to just say 'spatial and temporal autocorrelation'.*

Response: Thanks for your suggestion. We have revised it as follows:

"Utilizing spatial/temporal correlation is based on spatial and temporal autocorrelation, i.e., everything is related to everything else, but near things are more related than distant things."

**4. Results**

*L217: 'R2' -> 'R²' (here and throughout).*

Response: Thank you for pointing this out and we apologize for this point. We checked all mathematical symbols and corrected the incorrect ones.

*Figures 2 and 3: Could the ground reference and MODIS LAI uncertainties be shown as error bars?*

Response: We added the Figure 3 of accuracy comparison between two products and GBOV LAI under different vegetation types.

[Figure]

**Figure 3. Accuracy comparison between two products and GBOV LAI under different vegetation types. The numbers at the top represent the RMSE between the two products and the GBOV LAI reference, respectively.**

*Table 1: It might make more sense to report the statistics by biome, or at least show the biome on the table so the reader can see which sites belong to which biomes. Also, is there a reason why the bias and RRMSE are not shown here? What about the proportion of retrievals that were compliant with uncertainty requirements (e.g. those from GCOS - https://gcos.wmo.int/en/essential-climate-variables/lai/ecv-requirements or others).*

Response: Thanks for your suggestions, we added information about the biome type of site, the Relative RMSE (RRMSE), and Bias in Table 1.

**Table 1.** Comparison of MODIS LAI and HiQ-LAI over GBOV sites

| Biome Type | Site | M_RMSE | H_RMSE | M_R² | H_R² | M_RRMSE (%) | H_RRMSE (%) | M_Bias | H_Bias |
|---|---|---|---|---|---|---|---|---|---|
| | **CPER** | 0.44 | 0.40 | 0.21 | 0.22 | 75.90 | 69.32 | -0.39 | -0.37 |
| **Grasslands** | **KONA** | 1.26 | 1.19 | 0.47 | 0.68 | 102.32 | 96.31 | -1.20 | -1.16 |
| | **MOAB** | --- | --- | --- | --- | --- | --- | --- | --- |

| | | | | | | | | | |
|---|---|---|---|---|---|---|---|---|---|
| | ONAQ | 0.31 | 0.29 | 0.06 | 0.04 | 95.57 | 90.81 | -0.20 | -0.19 |
| | SRER | 0.21 | 0.21 | 0.84 | 0.86 | 45.16 | 44.94 | -0.11 | -0.05 |
| | STER | --- | --- | --- | --- | --- | --- | --- | --- |
| | WOOD | 1.29 | 1.19 | 0.55 | 0.66 | 110.19 | 102.07 | -1.24 | -1.15 |
| Forests | HARV | 0.82 | 0.53 | 0.72 | 0.83 | 19.37 | 12.41 | -0.24 | -0.15 |
| | TALL | 1.29 | 1.14 | 0.77 | 0.81 | 38.72 | 34.10 | -1.08 | -0.96 |
| | TUMB | 1.33 | 1.31 | 0.94 | 0.95 | 77.58 | 76.23 | -1.19 | -1.19 |
| Grasses & Shrubs | JORN | 0.06 | 0.16 | 1.00 | 1.00 | 14.95 | 35.91 | -0.03 | 0.01 |
| | VASN | --- | --- | --- | --- | --- | --- | --- | --- |
| Crops & Savannas | BLAN | 0.53 | 0.34 | 0.41 | 0.71 | 36.72 | 23.76 | -0.44 | -0.31 |
| | LAJA | 0.32 | 0.17 | 0.99 | 0.20 | 25.31 | 13.42 | -0.24 | -0.13 |
| Grasses & Savannas | GUAN | 0.72 | 0.49 | 0.36 | 0.43 | 22.23 | 15.13 | -0.59 | -0.33 |
| | JERC | 0.72 | 0.80 | 0.70 | 0.81 | 24.69 | 27.42 | 0.61 | 0.76 |
| | LITC | 0.85 | 0.68 | 0.58 | 0.66 | 149.71 | 120.21 | -0.83 | -0.66 |
| | NIWO | 0.49 | 0.31 | 0.68 | 0.64 | 66.89 | 42.19 | -0.47 | -0.30 |
| Savannas & Forests | BART | 0.75 | 0.46 | 0.80 | 0.89 | 19.60 | 11.87 | -0.14 | 0.06 |
| | DELA | 1.16 | 1.10 | 0.05 | 0.03 | 26.67 | 25.17 | 0.42 | 0.91 |
| | DSNY | 0.76 | 0.89 | 0.75 | 0.86 | 30.34 | 35.54 | 0.75 | 0.89 |
| | HAIN | 1.11 | 0.45 | 0.50 | 0.88 | 24.83 | 10.16 | 0.16 | 0.21 |
| | ORNL | 0.73 | 0.68 | 0.44 | 0.55 | 18.50 | 17.05 | -0.02 | 0.39 |
| | OSBS | 0.45 | 0.42 | 0.85 | 0.74 | 18.44 | 17.24 | 0.20 | 0.30 |
| | SCBI | 1.05 | 0.81 | 0.52 | 0.78 | 23.00 | 17.67 | 0.41 | 0.54 |
| | SERC | 0.84 | 1.38 | 0.83 | 0.80 | 18.85 | 30.92 | 0.75 | 1.31 |
| | STEI | 0.75 | 0.65 | 0.54 | 0.67 | 18.17 | 15.81 | -0.20 | 0.42 |
| | UNDE | 0.42 | 0.28 | 0.03 | 0.47 | 9.50 | 6.40 | 0.07 | 0.14 |
| | WOMB | --- | --- | --- | --- | --- | --- | --- | --- |

*L264-266: References needed here?*

Response: Thanks for your comments and we added the references as suggested.

"These regions are affected by the long-term influence of large cloud cover, high concentration of aerosol and saturation of Red-NIR, resulting in limited availability of high-quality observation data

and poor accuracy in LAI retrieval (Xu et al., 2018; Yan et al., 2016)."

**5. Discussion**

*L305: 'unusually high noise' – unusual compared to what?*

Response: Thanks for your comments, and we have revised it as follows:

"When affected by atmospheric conditions, sensor malfunctions, and retrieval algorithm uncertainties, MODIS LAI experiences poor spatiotemporal consistency and accompanies increased noise level."

*L325: Rather than being about 'the dense vegetation period of summer', I suspect this is more related to your next point – the high LAI is mostly found in tropical regions where there is greater cloud cover and aerosol load.*

Response: Thanks for your comments, and we have revised it as follows:

"Ground verification also reveals a consistent pattern between low MQA values and high LAI retrievals. Regions with high LAI values are mainly observed in tropical regions where there is greater cloud cover and aerosol load, which is prone to signal saturation."

*L326: Whilst I know what is meant by 'pixel quality is assessed based on the algorithm path' (i.e. main vs. back-up algorithm), it might not be clear to readers who are less familiar with the ins and outs of MODIS LAI. It would also be helpful to explain how this is assessed (I am assuming you are assigning a higher quality to the main algorithm and a lower one to the backup algorithm, but this is not explicitly mentioned).*

Response: Thank you for pointing this out, and we added the detail description in Sect 3.1.

"To address this issue, this approach employed multiple indicators to evaluate the uncertainty for each pixel (referred to as MQA hereafter). These indicators encompass the algorithm path, STD LAI, and Relative Time-series Stability (TSS). The algorithm path (AP) is a crucial quality index, distinguishing between the main and backup algorithms. The main algorithm offers superior quality and precision retrieval, and the weight ratio of the main algorithm and backup algorithm is determined as 6:4 in the previous study (Wang et al., 2023). STD LAI reflects the retrieval uncertainty. The AP and STD LAI are derived from the FparLai_QC and LaiStdDev layers of the original MODIS data. The third indicator, Relative TSS (RE-TSS), indicates the fluctuation of a time series (Zou et al., 2022). Following the principle of assigning a higher weight to smaller values, STD LAI and RE-TSS are incorporated into the retrieval with the main algorithm, resulting in the generation of a new quality classification indicator, MQA."

*L336: 'Equator' -> 'Equatorial' (here and throughout)?*

Response: Thanks for your comments, and we have replaced word of 'Equator'' in this paper with 'Equatorial''.

**References:**

[revised manuscript text omitted]

---

## Author Comment (AC4)

**Response to Reviewer:**

(Retype your comments in *italic font* and then present our responses to the comments)

*Thanks to the authors for considering my comments, which have mostly been addressed well.*

Response:Thank you so much again for providing positive feedback and detailed suggestions to improve our paper. We have carefully revised our manuscript, and adequately addressed all the questions and concerns that the referees have raised. Our responses are given below.

*I have three remaining concerns:*

*1. The 5 km product is now derived from the 500 m product using bicubic interpolation (i.e. cubic convolution) rather than nearest neighbour resampling. Whilst clearly an improvement, this is still problematic: bicubic interpolation takes account of a 4 x 4 window of pixels, which at 500 m, is equal to 2 km x 2 km. So, there is still a large discrepancy here. What is needed is to aggregate the pixels by calculating the mean of a 10 x 10 (= 5 km x 5 km) window (i.e. mean value downsampling). Then, nearest neighbour resampling can be carried out on this aggregated dataset to reproject it into whatever grid is required. As I mentioned in my last review, Google Earth Engine already has a function to accomplish this (i.e. the 'reduceResolution' method – please see under the 'Reduce Resolution' heading of https://developers.google.com/earth-engine/guides/resample).*

Response: Thanks for your comments again. We carefully considered your suggestion. We aggregate the pixels by calculating the mean of a 5 km x 5 km window. Then, nearest neighbour resampling can be carried out on this aggregated dataset to reproject it into the spatial resolution of 5km. The final resulting 5km dataset has been successfully stored in GEE (includes one LAI layer) and is accessible via the following link https://code.earthengine.google.com/?asset=users/JR_W/wgs_5km_8d_NeaNei.

*2. The authors have addressed my comment about PAI vs. LAI to some extent, in stating that 'since in-situ measurements may be sensitive to all elements of the canopy, the resulting estimate should technically be called the term plant area index'. However, more detailed discussion on how this influences the results is warranted. It's thought that woody material accounts for up to 35% of total plant area in forests (https://doi.org/10.1016/S0034-4257(99)00056-5), and recent work has shown that PAI may overestimate LAI by as much as 61% (https://doi.org/10.1016/j.ecoinf.2023.102441). Because of this, the conclusions drawn from assuming that PAI = LAI could easily be incorrect and influence interpretations on how well the products are performing. Since you are most interested in the relative improvement over the MODIS product, rather than the absolute validation results, the issue is not a major one (both products are being evaluated against the same reference data, and we can assume PAI is well correlated to LAI), but it does require some discussion. Additionally, I am of the opinion that we should call a spade a spade (and so it would be better for PAI to labelled PAI throughout to make it clear to the readers and avoid confusion), but this is obviously at the author's discretion.*

Response: Thanks for your suggestion again. We are primarily concerned that presenting LAI as PAI could lead to confusion among readers. Considering that most articles utilizing the GBOV site for verification refer to it as LAI, we have decided, through our discussions, to maintain this terminology without making any changes.

*3. It's written that 'the Inverse Distance Weighting (IDW) method is utilized on the spatial scale to calculate the weighted average of all eligible pixels (belonging to the same land cover type) within a certain spatial range of the target'. What is the 'certain spatial range'?*

Response: The 'certain spatial range' meaning is a spatial window size. The size of this spatial range is determined by calculating the size of the RMSE in previous studies (Wang et al., 2023).

In the algorithm for employing spatial information, the power exponent $\alpha$ and the half-width of the search window can control the influence of surrounding points on the interpolated target point and determine the utilization of the spatial information, respectively, and further affect the calculation results. In general, the smaller the half-width of the search window, the stronger the spatial correlation is, but the less spatial information is available. The larger the half-width of the search window, the more the spatial information can be employed, but the weaker the spatial correlation is. A higher power exponent results in less influence from distant points. Therefore, considering the balance between computational efficiency and accuracy, the optimal size of a half-width of the search window and a power exponent can be obtained. Fig. 12 shows the RMSE by varying the power exponent and the half-width of the search window. When the power exponent is equal to two, the gray line is essentially below the other lines. If the power exponent is constant, the RMSE decreases at first and then increases as the half-width of the search window increases. Hence, we determined the power exponent of 2 and the half-width of 4 pixels in our experiments.

[Figure]

Fig. 12. (a) $R\bar{M}SE$ was calculated from the average of RMSE in the spatial distribution. (b) Spatial distribution of RMSE by changing the correlation coefficient (CC, power exponent $\alpha$) and the half-width of the search window. The power exponent $\alpha$ controls the weight decay rate of the candidate pixels, and the half-width of the search window is the size of the neighborhood centered on the target pixel. We use power exponent $\alpha$ of 2 as the power exponent and 4 pixels for half-width in the algorithm.

In order to avoid the reader's misunderstanding, we have revised these sentences:

"Subsequently, the Inverse Distance Weighting (IDW) method is utilized on the spatial scale to calculate the weighted average of all eligible pixels (belonging to the same land cover type) within the half-width of 4 pixels and the power exponent of 2 (Wang et al., 2023) of the target pixel."

"On the temporal scale, the Simple Exponential Smoothing (SES) method is employed to calculate the weighted average of all eligible pixels within the smoothing parameter of 0.5 and the half-length of 3 (Wang et al., 2023)."

**References:**

[1]. Wang, J., Yan, K., Gao, S., Pu, J., Liu, J., Park, T., Bi, J., Maeda, E. E., Heiskanen, J., Knyazikhin, Y., and Myneni, R. B.: Improving the Quality of MODIS LAI Products by Exploiting Spatiotemporal Correlation Information, IEEE Trans. Geosci. Remote Sens., 61, https://doi.org/10.1109/TGRS.2023.3264280, 2023.

---

## Author Comment (AC5)

**Response to Reviewer:**

(Retype your comments in *italic font* and then present our responses to the comments)

*The paper introduces the High-Quality LAI (HiQ-LAI) dataset, which is an upgraded version of the existing MODIS LAI retrievals that typically suffer from high noise levels. The Spatio-Temporal Information Compositing Algorithm (STICA) was used to create this new dataset, which incorporates pixel quality information, spatio-temporal correlation, and original retrieval to provide more accurate results. The Time-series Stability (TSS) index showed that the area with smooth LAI time-series expanded significantly across the globe, especially in equatorial regions that are known to pose challenges for optical remote sensing. The HiQ-LAI dataset outperforms raw MODIS LAI in terms of continuity and consistency, both spatially and temporally, allowing for better land surface process simulation, climate modelling, and global change research. Overall, the manuscript is well-structured, provides valuable insights into vegetation dynamics across the globe, and has the potential to enhance the quality and impact of this field of research.*

*However, the advantages of the proposed method for solving the noise problem of the MODIS product are not adequately demonstrated in the paper. For instance, the authors state that the previous study "overlooked spatial correlation information" and that "genuine land surface LAI anomalies (e.g., caused by forest fires) may be artificially removed, even if the LAI profile appears smoother." The paper does not provide a clear explanation of how the proposed method addresses these issues.*

*The methods used in this study are straightforward, but some essential points are missing in the manuscript. For instance, a brief introduction of the STICA method, the TSS metric, and some key validation methods can further improve the clarity of the manuscript.*

Response: Special thanks for your positive comments and very detailed suggestions to make the paper better. Following the Reviewers' comments, we have carefully revised our manuscript, and adequately addressed all the questions and concerns that the referees have raised. Hope this revised manuscript has solved all your concerns.

In our algorithm, the quality, spatiotemporal information, and relative original observation records are fully utilized, and these pieces of information are weighted and averaged according to our fusion strategy. More robust results are obtained by considering multiple dimensions of information to compensate for the limitations of using a single information source and by preserving as real LAI anomalies as possible. Firstly, we introduce a novel quality evaluation index, MQA, to assess the quality of each pixel. The incorporation of temporal and spatial correlation also incorporates information from MQA, where the contribution of a pixel relies not only on its spatial/temporal correlation but also on its MQA value. Ultimately, the three dimensions of temporal, spatial, and the original observational value are integrated and furthest utilize the information from the original data. We anticipate that this process maximizes the preservation of genuine anomalies.

Additionally, we added a detailed description of the STICA method, the TSS metric, and some key validation methods in Sect 3.1 (red font)

[revised manuscript text omitted]

*Minor comments:*

*1. Line 21: Not entirely true. See Yuan et al., for instance.*

*(Yuan Hua; Dai Yongjiu; Xiao Zhiqiang; Ji Duoying and Shangguan Wei; Reprocessing the MODIS Leaf Area Index products for land surface and climate modelling, Remote Sensing of Environment, 2011, 115(5): 1171-1187.)*

> Response: Thanks for your comments and we had overwritten this sentence as follows:
>
> "Reprocessing MODIS LAI predominantly rely on temporal information to achieve smoother LAI profiles with little use of spatial information and may easily ignore genuine LAI anomalies."

*2. Line 67: Please check the sentence.*

> Response: Thanks for your comments and we had overwritten this sentence as follows:
>
> "The best retrievals are then selected using the temporal compositing method, and the 4-day or 8-day product is generated from the daily retrievals. Therefore, MODIS LAI retrievals are calculated independently for each pixel and daily. Differences in adjacent observation conditions lead to significant uncertainty in the LAI time series."

*Line 85: The author may need to justify how the previous study "overlooked spatial correlation information."*

> Response: Maybe there is a certain misunderstanding in our expression. We had overwritten this sentence as follows: "While these methods effectively utilize temporal and QC layers information, they frequently overlook the utilization of spatial information or rely on spatial correlation as an alternative and place a greater emphasis on leveraging temporal information."

*Line 197: Why did the authors choose the year 2021? Have they tested other years, such as 2015, when strong ocean modes occurred, and NH vegetation growth was notably affected?*

*(Bastos Ana; Ciais Philippe; Park Taejin; Zscheischler Jakob; Yue Chao; Barichivich Jonathan; Myneni Ranga B.; Peng Shushi; Piao Shilong and Zhu Zaichun; Was the extreme Northern Hemisphere greening in 2015 predictable?, Environmental Research Letters, 2017, 12(4): 044016.)*

> Response: We chose the BELMANIP V2.1 site to assess the global consistency between HiQ-LAI and MODIS LAI across various vegetation types in 2021 to identify vegetation types that exhibit greater consistency as well as those that diverge the most. For this purpose, we considered the Reviewer's suggestion and conducted a comparison of scatterplots representing the two products in 2015. The findings indicated that the distinction between the two products in 2015 remained relatively consistent with that observed in 2021.

[Figure]

**Figure. Density scatter plots comparison of MODIS LAI and HiQ-LAI in 2015.**

[Figure]

**Figure 5. Density scatter plots comparison of MODIS LAI and HiQ-LAI in 2021 using the BELMANIP V2.1 sites (445 sites). B1: grass and cereal crops, B2: shrub, B3: broadleaf crops, B4: savanna, B5: evergreen broadleaf forest, B6: deciduous broadleaf forest, B7: evergreen coniferous forest, B8: deciduous coniferous forest.**

*Line 240: R2 should be corrected. Please check the entire manuscript. For example, see also Line 261 and 263.*

Response:Thank you for pointing this out and we apologize for this point. We have checked all

mathematical symbols and corrected the incorrect ones.

"The result demonstrates that, except for B5, the $R^2$ for other pure vegetation types exceeds 0.88, and B1 and B3 surpassed 0.95. The consistency of mixed pixels is also relatively high, as indicated by an RMSE of 0.42 and an $R^2$ of 0.86. However, B5 exhibits a significant disparity, with an $R^2$ value of 0.15."

"In the Poor-Quality level, HiQ-LAI exhibited a 17.81% increase in $R^2$ and an 18.99% reduction in RMSE compared to MODIS LAI."

*Line 386: Please rephrase the sentence.*

Response: Thanks for your comments, and we have deleted this sentence (According to another reviewer's comments).

*Figure 7: Could the authors explain the presence of the unusual stripes observed in the northern high latitudes? Additionally, providing a difference map between the two would offer a more straightforward visual representation. Finally, the histogram indicates that the colors in the colorbar were not effectively used.*

Response: Since the original MODIS image is missing the data of the entire latitude band in the high latitude region from December to January, this may be the reason for the obvious banding in the high latitude region. Additionally, we added the difference chart between the two products in Fig. 9 and added the color percentage values representing the top two in terms of the proportion of LAI trend value.

[Figure]

**Figure 9. Global maps of LAI trends between MODIS LAI (a) and HiQ-LAI (b) during 2000 − 2022. The Theil–Sen's slope (TS) method and Mann-Kendall (MK) test were used to calculate these results. (c) Difference of LAI trends between MODIS LAI and HiQ-LAI.**

*Figure 9: The authors should provide more details on how the "improvement percentage of RMSE and R2" was calculated.*

Response: Thanks for your suggestions, and we added a detailed description in Sect 3.2 of how the "improvement percentage of RMSE and R2" was calculated.

"we used DIRECT V2.1 ground measurements in this research (Morisette et al., 2006; Garrigues et al., 2008). However, these data were not utilized for direct validation due to the discontinuity in the observed time series at these sites. Instead, the DIRECT V2.1 sites provided valuable reference values in Sect. 5.2. Similarly, a research area of 6 × 6 pixels was selected for each site, and we compared the R² and RMSE of the two products with sites across different quality grades. The analysis involved determining the RMSE reduction percentage and R² increase in the percentage of HiQ-LAI relative to MODIS under various quality grades."

*Figure 10: Please label the LAI units for the corresponding colorbars.*

Response: Thanks for your comments, we added the LAI units in Fig. 12 and Fig. 13.

[Figure]

**Figure 12. Spatial distribution of MODIS LAI (a1-a4) and HiQ-LAI (b1-b4) in equatorial region within different composite day.**

[Figure]

**Figure 13. Spatial distribution of MODIS LAI (a1-a4) and MQA (b1-b4) values over equatorial region.**